# ML-Bench: Evaluating Large Language Models and Agents for Machine Learning Tasks on Repository-Level Code

## Abstract

Despite Large Language Models (LLMs) achieving impressive results in code generation, significant challenges remain in automated ML development, particularly in utilizing existing ML repositories effectively. Also, recently, people have developed LLM agents that attempt to interact with repository code (e.g., resolving issues), prompting the need for end-to-end evaluations starting from environment setup to deploying the repository rather than merely generating code in already-configured environments. These two gaps have motivated our development of ML-Bench, a benchmark rooted in real-world ML applications that leverage existing code repositories. ML-Bench encompasses annotated **9,641** examples across **18** GitHub repositories, challenging LLMs to accommodate user-specified arguments and documentation intricacies effectively. To evaluate both LLMs and agents, two setups are employed: ML-Bench-L for assessing LLMs' text-to-code conversion within a predefined deployment environment, and ML-Bench-A for testing autonomous agents in an end-to-end task execution within a Linux sandbox environment. Our findings indicate that while GPT-4o leads with a Pass@5 rate surpassing **50%**, there remains significant scope for improvement, highlighted by issues such as hallucinated outputs and difficulties with bash script generation. Notably, in the more demanding ML-Bench-A, GPT-4o achieves a **76.47%** success rate, reflecting the efficacy of iterative action and feedback in complex task resolution. Our code is available at `https://anonymous.4open.science/r/ML-Bench` and our data is in the supplementary material.

## 1 Introduction

Large Language Models (LLMs) have demonstrated remarkable prowess in function-level code generation (Austin et al., 2021; Chen et al., 2021; Hendrycks et al., 2021b; Li et al., 2022). Recent benchmarks have shifted from simple function synthesis to more complex tasks such as code editing and debugging (Cassano et al., 2023; Tian et al., 2024; Haque et al., 2023; Li et al., 2024) and coding within a repository context (Ding et al., 2024; Zhang et al., 2023a; Li et al., 2024; Yu et al., 2024). Furthermore, the evolution of code generation benchmarks reflects a growing recognition of the need for more realistic evaluation scenarios (Guo et al., 2024), like proficiency with data science libraries (Lai et al., 2023; Ma et al., 2024), programming with external tools and APIs (Li et al., 2023; Shen et al., 2023; Wang et al., 2023a; Gao et al., 2024).

While benchmarks like SWE-bench (Jimenez et al., 2024) have established strong foundations for evaluating repository-level code understanding, and MLAgentBench (Huang et al., 2023) has highlighted the importance of ML capabilities, a critical gap remains in evaluating models' ability to utilize existing ML repositories correctly. Rather than testing models' capability to implement ML algorithms from scratch, we focus specifically on how well models can understand and execute workflows using established ML codebases - a crucial skill for practical ML development. This gap is particularly significant given the recent surge of research in LLM-based agents for data science and ML tasks (Hong et al., 2024; Hassan et al., 2023).

We introduce ML-Bench based on common real-world ML workflows, often using existing ML repositories as libraries, as shown in Figure 1. To better assess the abilities of LLMs and agents at

the same time, we present two testing setups: ML-BENCH-L and ML-BENCH-A; examples can be found in Figure 2 and Appendix E:

- ML-BENCH-L: Evaluates models' capacity to complete tasks within a *pre-configured deployment environment*, translating text instructions to simple bash or Python code with clearly defined parameters. The environment is already set up with the necessary dependencies and datasets.

- ML-BENCH-A: Introduces a secure Linux sandbox environment where agents start with an *empty Docker container* and must iteratively execute commands and code blocks to set up the environment, install dependencies, download datasets, and finally execute the task, emulating the full workflow of a human coder.

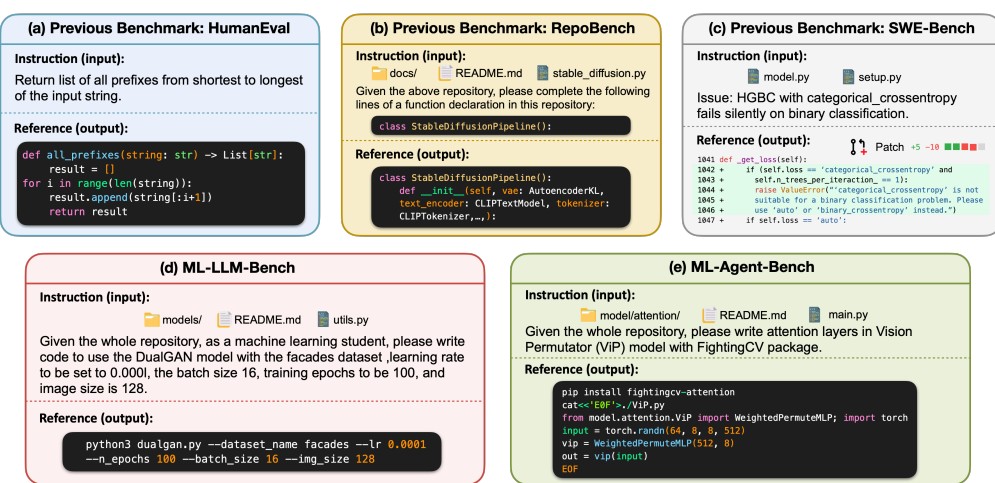

Figure 1: Examples of ML-BENCH compared with existing code benchmarks HumanEval (Chen et al., 2021), RepoBench (Liu et al., 2023), and SWE-bench (Jimenez et al., 2024). In ML-BENCH, (1) models must **take repository-level code as input**, and (2) based on their understanding of the repository, models are required to compose **new code segments that do not exist** within the original repository.

In contrast to some focused efforts in developing LLM agents for ML tasks, such as feature engineering (Hollmann et al., 2024), hyper-parameter tuning (Zhang et al., 2023b), aiding AI research (Huang et al., 2023), and data operations (Lai et al., 2023), ML-Bench takes a broader approach. Our work advances this by enabling agents to not only execute ML experiments but also automatically configure and set up repositories. The novelty and contributions of ML-Bench are: (1) We specifically evaluate models' ability to automate complex ML workflows, including environment setup, dependency management, and experiment execution. (2) Our four distinct evaluation settings provide insights into models' true capabilities while addressing data leakage concerns.

ML-BENCH-L benchmarks their competence in *translating text instructions to simple bash code with clearly defined parameters*. It seeks to test whether LLMs can generate executable code to invoke specific files or functions in a repository with appropriate arguments based on given instructions. For instance, it might assess if an LLM can generate a command line to utilize `txt2img.py` from an image generation model repository with parameters such as `ckpt` to produce an image based on a text description, e.g. `python txt2img.py --prompt "a girl riding a horse" --ckpt SD2_1_v_model.ckpt`. To address this, LLMs must understand the repository-level code and accurately configure parameters. Another critical aspect of this process is understanding documentation—especially README files—which typically include comprehensive instructions on employing the library, complete with task examples and argument selection guidelines.

However, a more arduous challenge lies in the end-to-end execution of tasks, starting from scratch. This involves initiating the code environment for a specific repository, where common pitfalls of environment setup, such as missing datasets or uninstalled packages, might occur. To evaluate agents in such a setup, we introduce ML-BENCH-A, which provides a secure Linux sandbox environment where agents can *iteratively execute commands and code blocks to obtain feedback*. The agent's actions involve multiple attempts, from reading files and understanding the repository to installing

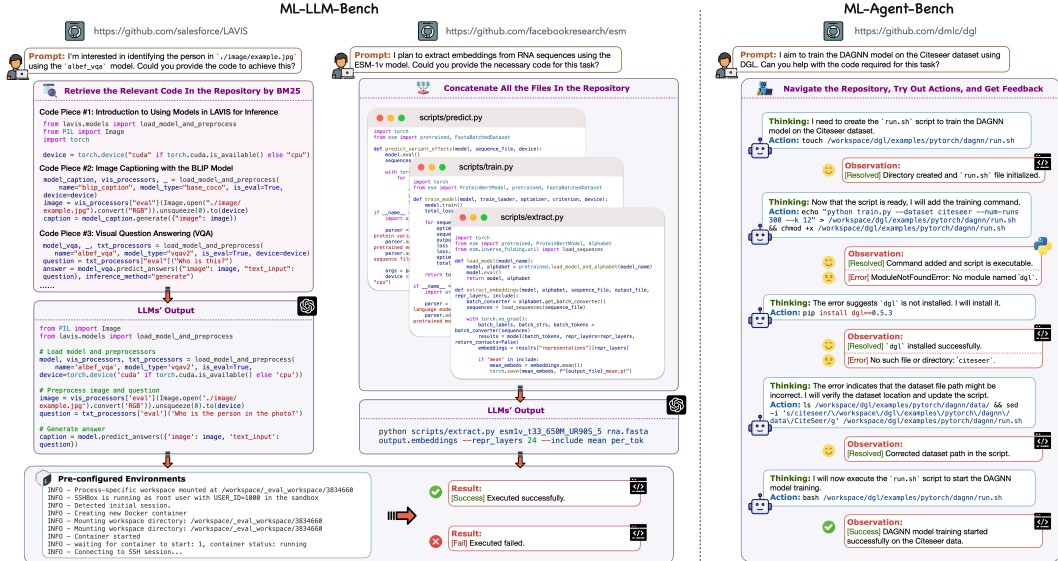

Figure 2: The workflow of ML-BENCH, including ML-Bench-L and ML-Bench-A. In ML-Bench-L, LLMs generate Python code or Bash scripts based on the prompt. The input to the LLMs could be code retrieved from a repository based on the prompt or a direct concatenation of all files. Their performance is evaluated within a pre-configured environment. Conversely, in ML-Bench-A, the agent must autonomously set up the environment and download necessary datasets to accomplish the task.

dependencies, preparing datasets, and finally writing bash code that calls the repository, thus emulating the full workflow of a human coder.

ML-BENCH features **9,641** samples from **18** ML GitHub repositories, as Figure 3. In our evaluation experiment on ML-BENCH-L, we observe that GPT-4o outperforms other LLMs, being the sole model to surpass the **50%** threshold in the Pass@5 metric (success rate within five tries). It is noteworthy that in the same test set, our annotators' performance—computer science graduate students—stood at a success rate of **86.76%**, with **59** out of **68** examples correctly executed. This indicates substantial room for improvement in current LLMs. However, the models did show performance improvements following instruction tuning on the training data (**8.85**→**15.76** for CodeLlama). Error analysis reveals that LLMs tend to generate hallucinations, predominantly producing incorrect parameters or referencing non-existent files. Generating

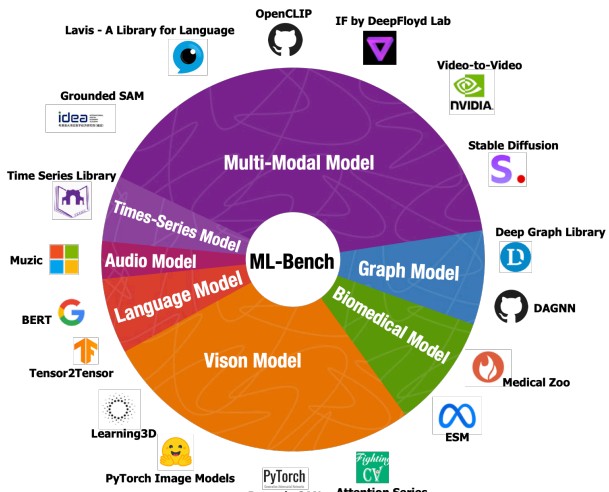

Figure 3: ML-BENCH ENCOMPASSES **18 PROMINENT GITHUB REPOSITORIES** AND IT SHOW THE DISTRIBUTION OF **9,641 SAMPLES**.

bash scripts proved more challenging than generating Python code, pinpointing a capability bottleneck in LLMs. A critical insight from our study is the urgent need for LLMs to comprehend the long code context (the average length is around 150k tokens for the whole repository), not merely to generate code. On the more challenging ML-BENCH-A setup, GPT-4o scores **76.47%** within the OpenDevin agent environment, where agents must configure their environment, navigate code repositories, and effectively generate the necessary code. This underscores the potential of self-improvement and incorporating feedback from experience as alternatives to relying on instruction tuning with history training data to enhance LLM performance.

Table 1: Comparison of benchmarks for repository-level code analysis: this comparison focuses on several key attributes across various benchmarks: (1) *Repository Understanding*—the ability to comprehend and navigate the overall structure, dependencies, and functionality of an entire code repository beyond individual files; (2) *Documentation Understanding*—the capability to interpret and utilize documentation elements such as README files to gain insights within the repository; (3) *Cross-File Retrieval*-identifying relevant information across multiple files to complete tasks or resolve issues; (4) *Package Installation*—installing dependencies required for the repository; (5) *Data Downloading*—downloading data required for the task; and (6) *Evalution*-the methods used to assess and measure the task performance.

| Criteria | REPOEVAL (Zhang et al., 2023a) | REPOBENCH (Liu et al., 2023) | MLAGENTBENCH (Huang et al., 2024) | SWE-BENCH (Jimenez et al., 2024) | ML-BENCH (OURS) |
|---|---|---|---|---|---|
| Repo. Understanding | ✓ | ✓ | ✗ | ✓ | ✓ |
| Doc. Understanding | ✗ | ✗ | ✓ | ✗ | ✓ |
| Cross-File Retrieval | ✗ | ✓ | ✓ | ✓ | ✓ |
| Package Installation | ✗ | ✗ | ✗ | ✗ | ✓ |
| Data Downloading | ✗ | ✗ | ✗ | ✗ | ✓ |
| Evaluation | Similarity / Pass@K | Similarity | Test Accuracy | Success Rate | Pass@K / Success Rate |
| # of Repositories | 14 | 3,116 | / | 12 | 18 |
| # of Tasks | 13,710 | 49,684 | 13 | 2,300 | 9,641 |

To sum up, while recent efforts have explored LLM-based agents for navigating GitHub repositories, such as conducting ML experiments in simplified environments (Huang et al., 2024) or resolving repository issues (Jimenez et al., 2024) (see Table 1), ML-Bench addresses a distinct and critical challenge faced by many machine-learning researchers: both *setting up* and *executing experiments* using research repositories in-the-wild. Compared to existing work, our contributions are:

- SWE-Bench (Jimenez et al., 2024) tasks agents with locating and modifying specific functions to resolve an issue within a pre-deployed testing environment. ML-Bench challenges agents to independently configure environments and download necessary data, mimicking real-world research scenarios more closely.

- While MLAgentBench (Huang et al., 2023) evaluates LLMs' ability to run simple ML experiments, it focuses on optimizing ML experiments rather than comprehending and setting up a repository for experimentation. ML-Bench goes beyond this by requiring agents to utilize machine-learning codebases.

- ML-Bench evaluates the entire workflow of (1) setting up, e.g., downloading/installing existing datasets, models, & packages, and (2) running ML experiments, from initial repository exploration to result in interpretation. We have released a one-click evaluation code, facilitating easy use and extension of the benchmark by researchers.

## 2 ML-BENCH CONSTRUCTION

### 2.1 TASK FORMULATION AND DESIGN PRINCIPLE

ML-BENCH aims to test models' ability to utilize existing ML repositories according to user requirements. For each task, a model receives a GitHub repository, a natural language instruction, and specific parameter requirements. The model must then generate executable code that correctly uses the repository's functions or models while adhering to the provided requirements.

For ML-BENCH settings, (1) ML-BENCH-L provides a complete build environment, allowing us to test output bash scripts generated by LLMs within a Docker environment regarding the correctness and excitability. (2) ML-BENCH-A provides agents with access to an empty Docker environment without essential packages. Agents must attempt to download the requirements for each user instruction involving the installation of new datasets or Python packages themselves. This design ensures that our testing framework aligns with practical application workflow.

ML-BENCH focuses on the end-to-end task of setting up and executing research-related tasks in repositories, presenting a unique set of challenges not fully addressed by existing benchmarks: a) Models must comprehend both code and README files to navigate complex ML repositories. b) Tasks require comprehension, use, modification, and reasoning across multiple files within a repository. c) Agents must configure environments, install dependencies, download datasets, and acquire necessary models. d) Agents need to make sequential decisions while interacting with the environment, mimicking real-world research scenarios. e) Unlike previous work that typically

supports either system shell commands (Yang et al., 2024) or Python commands (Huang et al., 2023), ML-Bench provides an environment allowing both (this resembles a scientist's workflow to interact with both environments). Each execution is equivalent to running a cell containing Python code and/or bash commands, with state preserved between cell executions.

Table 2: Detailed breakdown of the number of bash script and Python code samples for each repository. The test set contains samples from **14** repositories, while the train set includes **4** additional repositories for the OOD setting. A quarter subset of the test set is also shown. All repository names are hyperlinked for direct access to the corresponding GitHub.

| Repository | Train Set | | Test Set | | 1/4 Test Set | |
|---|---|---|---|---|---|---|
| | Scripts | Code | Scripts | Code | Scripts | Code |
| **In-Distribution (ID)** | | | | | | |
| Video-to-Video (vid2vid) | 46 | 0 | 13 | 0 | 4 | 0 |
| IF by DeepFloyd Lab (If) | 168 | 175 | 10 | 11 | 4 | 2 |
| Deep Graph Library (DGL) | 553 | 0 | 21 | 0 | 5 | 0 |
| Pytorch-GAN (Py-GAN) | 1080 | 0 | 30 | 0 | 8 | 0 |
| ESM | 563 | 58 | 15 | 2 | 4 | 1 |
| BERT | 962 | 0 | 22 | 0 | 6 | 0 |
| OpenCLIP | 646 | 691 | 10 | 1 | 3 | 0 |
| Lavis - A Library for Language (Lavis) | 76 | 205 | 4 | 23 | 1 | 6 |
| Time Series Library (TSL) | 1449 | 0 | 14 | 0 | 4 | 0 |
| Attention Series (EAP) | 95 | 5 | 24 | 0 | 5 | 0 |
| **Out-Of-Distribution (OOD)** | | | | | | |
| Grounded-SAM | / | / | 12 | 8 | 2 | 3 |
| PyTorch Image Models (Py-IM) | / | / | 5 | 0 | 1 | 0 |
| muzic | / | / | 17 | 1 | 4 | 1 |
| Learning3D | / | / | 17 | 0 | 4 | 0 |
| Stable Diffusion (SD) | 2253 | 0 | / | / | / | / |
| Medical Zoo (MedZooPy) | 490 | 0 | / | / | / | / |
| Time Series Library (TCL) | 196 | 0 | / | / | / | / |
| Tensor2Tensor | 0 | 248 | / | / | / | / |
| Total | **8577** | **736** | **214** | **46** | **55** | **13** |

## 2.2 Summary of Data

ML-Bench contains 18 diverse repositories, each reflecting varying complexity and tasks, while filtering out substandard samples. The data quantities and breakdown per repository are detailed in Table 2. Regarding the code language, our annotated output includes both **bash scripts**, which invoke Python files with specific arguments, and **Python code**, which calls functions from the repository. Bash scripts significantly outnumbered Python code snippets (See Appendix A for the explanation).

Each repository contributed approximately 480 examples, summing up to 9,641 examples. For our experiments involving the fine-tuning of open-source LLMs, we split the dataset based on code origin: The **In-Distribution (ID)** approach utilizes data from the same repository both for training and testing, allowing repository-specific code to be exposed to models during fine-tuning. In contrast, the **Out-Of-Distribution (OOD)** method employs disjoint sets for training and testing, encompassing eight repositories—half for model training and the remaining for evaluation. The overall statistics and further detailed data metrics for each repository utilized can be found in Appendix G.4 and G.3.

## 2.3 Data Collection and Annotation Pipeline

Eight computer science graduate students with proficient programming abilities contributed to our data annotation, with each repository's related data being the responsibility of one annotator and an additional reviewer to ensure accuracy. These students, who are co-authors of this paper, brought their domain expertise to ensure high-quality annotations. Annotators were permitted to use GPT-4 to expedite the annotation, although manual verification and adjustments were required. Annotating a repository took approximately 5-10 hours (Appendix D). The annotation workflow is shown in Figure 4:

**(1) README file Selection**: Annotators commenced by meticulously reviewing repository contents to identify all README files, including those within various subdirectories, each covering different functionalities. On average, a GitHub repository included 12 README pages, with one notable repository, DGL, comprising 154 README files. **(2) Task Mining**: Annotators identify practical

Figure 4: A detailed construction pipeline of ML-BENCH.

tasks from README files, along with corresponding code examples, averaging nine tasks per repository, thus capturing the representative functionality of each GitHub repository. Annotators randomly selected 20-30 test set candidates after annotating each repository. These candidates underwent additional executable verification and correctness checks. Cross-validation was performed by other annotators to remove near-duplicate cases, ensuring that cases differing only in minor parameter values or paths but representing the same task were excluded. **(3) Arguments Extraction**: Beyond task identification, annotators with machine learning expertise extracted key parameters essential for task completion, targeting representative parameters commonly employed in practical experiments. **(4) Multi-Argument Combination**: With tasks and arguments identified, annotators create diverse combinations of argument values, essential for constructing scenarios that represent real-world applications of repository code. **(5) Instruction Generation**: Utilizing ChatGPT, we generate task-specific instructions for each argument set, applying templates to ensure diversity and explicit argument inclusion, detailed in Appendix G.2. **(6) Reference Code Generation**: For each argument combination, we develop code templates to generate accurate ground truth code for the targeted tasks. **(7) Quality Check**: The dataset underwent stringent quality checks, particularly for code executability and argument accuracy, with any non-compliant data being revised or discarded. This ensures that the instructions precisely align with user requirements, thereby upholding the integrity and applicability of the ML-BENCH benchmark. We conducted three additional quality assessments with human evaluation, and the details of quality control are included in Appendix G.5. In addition, We mitigate the risk of data contamination by manually rewriting inputs and outputs and verifying our dataset's uniqueness against internet searches.

## 3 ML-BENCH-L EXPERIMENTS

### 3.1 ML-BENCH-L SETUP

Our experimental inputs include human instructions and the entire repository code (including README files). We present three distinct experimental setups to evaluate the models. Given that current models cannot process the entire code context, the three scenarios range from ideal to extreme. **Oracle Segment (Oracle):** For the Oracle setup, annotators identify and record crucial segments within README files — referred to as "Oracle Segments" — that contain necessary codes and textual explanations pivotal for completing the prescribed tasks. These segments serve as the foundational source to derive the ground truth code, ensuring that models can access all critical evidence when generating code. **BM25 Retrieval (Retrieval):** In this setup, we employ a BM25 retriever to extract segments from the repository's documentation, including README files, that are relevant to the given instructions. This method aims to mimic a more automated way of narrowing down necessary information without human pre-selection. **Code (Code):** This setting exposes the model to the entire code repository. All files within the repository, with README files placed at the forefront, are presented as input to the model. Due to model context limitations, texts are truncated when necessary, and potential information loss is analyzed and documented in Appendix H.2. Please refer to Appendix H.1 for further details on implementing the BM25 retriever.

### 3.2 EVALUATION METRICS

The generated code must be executable and adhere to the parameters outlined in the user instructions. We use Pass@K as our metric for evaluation, with K representing the number of generation attempts allowed. Pass@K measures the likelihood of the model producing at least one correct code execution in those K tries (given unit tests).

Table 3: Pass@1/5 scores for models on the SCRIPTS|CODE (bash script, Python code) partition of ML-BENCH-L. †denotes instruction-tuned models. Results are shown for the **Oracle**, **Code**, and **Retrieval** settings. Results under ID and out-of-distribution (OOD) are reported after instruction fine-tuning. SCRIPTS|CODE superscript: numbers represent the breakdown of performance on bash script generation tasks (SCRIPTS) versus Python code generation tasks (CODE). The reported numbers are weighted averages of Scripts and Code scores.

| Models | Oracle SCRIPTS\|CODE | | Code SCRIPTS\|CODE | | Retrieval SCRIPTS\|CODE | |
|---|---|---|---|---|---|---|
| | **Pass@1** | **Pass@5** | **Pass@1** | **Pass@5** | **Pass@1** | **Pass@5** |
| Human | / | / | 86.76 | / | / | / |
| *Close-Source LLMs* | | | | | | |
| GPT-4o | $36.42^{31.37\|56.83}$ | $50.13^{44.26\|78.89}$ | $32.99^{31.44\|39.87}$ | $46.20^{43.58\|61.54}$ | $26.16^{19.47\|55.52}$ | $30.44^{24.73\|76.92}$ |
| GPT-4 | $33.82^{29.09\|53.85}$ | $48.53^{41.81\|76.92}$ | $30.88^{29.09\|38.46}$ | $45.59^{41.82\|61.54}$ | $22.06^{14.55\|53.85}$ | $27.94^{16.36\|76.92}$ |
| GPT-3.5 | $27.94^{21.81\|53.85}$ | $38.23^{30.91\|69.23}$ | $15.07^{0.09\|38.46}$ | $30.14^{23.64\|53.85}$ | $13.70^{5.45\|46.15}$ | $24.66^{14.55\|69.23}$ |
| Claude-3-Opus | $25.52^{12.15\|67.39}$ | $36.92^{27.57\|80.43}$ | $13.46^{0.70\|43.48}$ | $35.39^{30.37\|58.70}$ | $10.00^{3.27\|41.30}$ | $22.69^{11.22\|76.09}$ |
| Claude-3-Sonnet | $21.92^{18.18\|38.46}$ | $34.25^{27.27\|61.54}$ | $27.40^{25.45\|30.76}$ | $35.62^{30.91\|53.85}$ | $9.59^{3.64\|38.46}$ | $20.55^{9.09\|69.23}$ |
| Claude-3-Haiku | $18.46^{11.68\|50.00}$ | $30.38^{20.09\|78.26}$ | $25.38^{22.90\|36.96}$ | $32.31^{28.04\|52.17}$ | $8.08^{3.74\|28.26}$ | $16.92^{7.48\|60.87}$ |
| *Open-Source LLMs* | | | | | | |
| CodeLlama-7b | $8.85^{3.37\|32.60}$ | $21.15^{11.68\|65.22}$ | $1.54^{0.47\|6.52}$ | $8.85^{2.80\|36.96}$ | $0.77^{0.00\|4.34}$ | $8.85^{2.80\|36.96}$ |
| DeepseekCoder-6.7b | $9.23^{0.46\|30.43}$ | $24.23^{14.02\|71.74}$ | $3.85^{1.89\|13.04}$ | $10.38^{6.07\|30.43}$ | $5.00^{3.27\|13.04}$ | $14.23^{9.81\|34.78}$ |
| Llama-2-7b | $2.27^{0.13\|5.70}$ | $4.77^{2.47\|6.22}$ | 0.00 | 0.00 | 0.00 | 0.00 |
| Llama-3.1-8B | 32.69 | 37.31 | 12.31 | 13.85 | 16.54 | 22.69 |
| Llama-3.1-70B | 32.69 | 37.31 | 12.31 | 13.85 | 16.54 | 22.69 |
| Llama-3.1-405B | 15.38 | 33.85 | 13.46 | 23.85 | 4.23 | 10.38 |
| Deepseek-Chat-6.7b | 25.00 | 27.69 | 10.38 | 11.15 | 9.23 | 11.92 |
| DeepSeek-Coder-6.7b | 32.69 | 37.31 | 12.31 | 13.85 | 16.54 | 22.69 |
| Qwen2.5-7b | 33.46 | 47.31 | 12.31 | 18.08 | 11.92 | 19.38 |
| Qwen2.5-32B | 40.38 | 51.92 | 15.00 | 19.23 | 22.31 | 32.31 |
| Qwen2.5-72B | 38.08 | 47.69 | 17.31 | 20.38 | 12.69 | 21.54 |
| *Finetuned LLMs w/ the Out-Of-Distribution (OOD)* | | | | | | |
| CodeLlama-7b † | $15.76^{12.14\|32.61}$ | $28.46^{19.62\|69.57}$ | / | / | $1.92^{0.47\|8.70}$ | $5.38^{1.40\|23.91}$ |
| DeepseekCoder-6.7b † | $16.15^{14.95\|34.78}$ | $31.15^{24.30\|58.70}$ | / | / | $10.38^{6.54\|28.26}$ | $26.15^{17.29\|67.39}$ |
| Llama-2-7b † | $5.31^{2.47\|10.86}$ | $6.03^{3.12\|11.64}$ | / | / | $2.77^{1.30\|5.34}$ | $5.31^{2.47\|10.86}$ |
| *Finetuned LLMs w/ the In-Distribution (ID)* | | | | | | |
| CodeLlama-7b † | $17.69^{15.42\|28.26}$ | $30.77^{21.96\|71.74}$ | / | / | $2.69^{0.47\|13.04}$ | $9.62^{3.27\|39.13}$ |
| DeepseekCoder-6.7b † | $21.92^{12.16\|65.22}$ | $30.77^{20.56\|78.26}$ | / | / | $2.69^{1.40\|8.70}$ | $10.00^{5.61\|30.43}$ |
| Llama-2-7b † | $6.54^{2.33\|26.09}$ | $8.38^{4.45\|32.17}$ | / | / | $1.15^{0.00\|6.52}$ | $3.08^{4.67\|15.22}$ |

Table 4: Agent evaluation results on the ML-BENCH-A. The success rate, number of solved instances, and the average cost per solved instance are reported for each agent and language model combination. † Evaluation is conducted on a quarter subset of the test set due to budget constraints.

| Agent | Model Name | Success Rate[†] (%) | # of Solved Instances | $ Avg. Cost |
|---|---|---|---|---|
| AutoGen (Wu et al., 2023) | `gpt-4-1106-preview` | 8.82 | 6 | 1.28 |
| SWE-Agent (Yang et al., 2024) | `gpt-4-1106-preview` | 42.64 | 29 | 1.91 |
| Aider (Gauthier) | `gpt-4o` | 64.38 | 47 | - |
| OpenDevin (Wang et al., 2024b) | `gpt-4o-2024-05-13` | 76.47 | 51 | 0.25 |
| | `gpt-4-1106-preview` | 58.82 | 40 | 1.22 |
| | `gpt-3.5-turbo-16k-0613` | 13.23 | 9 | 0.12 |

## 3.3 EXPERIMENTAL RESULTS

As presented in Table 3, we conducted evaluations on a set of LLMs including GPT-4o (model name: `gpt-4o-2024-05-13`), GPT-4 (model name: `gpt-4-1106-preview`), GPT-3.5 (model name: `gpt-3.5-turbo-16k-0613`), and the Claude 3 model family (`Claude-3-Opus`, `Claude-3-Sonnet`, `Claude-3-Haiku`). Moreover, we selected `CodeLlama-7b-Instruct`, `DeepSeek-Coder-6.7b-Instruct`, and `Llama-2-7b-chat-hf` to explore the effects of fine-tuning with an 8k token length limit with 4 A100s. The findings suggest that while GPT-4o exhibited the highest scores across the test cases, the untrained models, such as `LLama-2-7b`, performed poorly on the ML-BENCH-L, even after in-distribution (ID) fine-tuning. Fine-tuning on out-of-distribution (OOD) data indicated that models could benefit from training on similar tasks, though not necessarily from the same repository. Moreover, the performances on ID data implied that even after task-relevant fine-tuning, the results from 7B-scale open-source models could not outperform the closed-source counterparts. The oracle setting outcomes demonstrate that providing models with the correct reference solutions is effective for task completion. A retrieval approach not specifically designed for the task might lead to suboptimal results, potentially hindering performance.

# 4 ML-BENCH-A EXPERIMENTS

## 4.1 ML-BENCH-A SETUP

In ML-BENCH-A, as shown in Figure 5, we provision a sandbox environment as the testing ground for agents. The sandbox offers a fundamental setup, such as a configurable Docker image, allowing agents to modify and execute commands freely within the simulation. Agents are granted the ability to execute bash scripts or interact with IPython notebooks. The agents must interact with this environment, perusing code within repositories—regardless of the extended code or required parameters—to accumulate comprehensive information. This process necessitates successive actions, with the agent autonomously determining the correctness of each step and iteratively refining its approach upon encountering errors. We expect the agents' outputs to differ from previous LLM settings due to the dynamic and interactive nature of the tasks.

Recent agent frameworks, including SWE-Agent (Yang et al., 2024), Aider (Gauthier), OpenDevin (Wang et al., 2024b), provide a well-defined suite of impactful actions that bridge the agent with its operational environment. Like Reflexion (Shinn et al., 2023) and CodeAct (Wang et al., 2024a), agents iteratively execute actions, refine their approach via feedback, and perform effectively in solving complex tasks. These agents are designed to mimic the workflow of human programmers, thoroughly parsing and employing a repository. To facilitate this, the agents can execute *any* Python code and bash commands within a secure and isolated Linux OS sandbox, providing an ideal setting for our benchmark evaluations. In each instance, ML-BENCH-A initiates an isolated docker container sandbox where all agents' bash commands are executed, with the outcomes returned as observations. Different agent frameworks implement environmental interactions in varying ways, with each action yielding observations for AI agents. Here ML-BENCH-A essentially assesses the effectiveness of different environments. In ML-BENCH-A, a configurable workspace directory contains repositories agents are to handle, installed within a safe sandbox environment that provides controlled access for agents to interact with and process as needed. For evaluation, instead of relying on the Pass@K metric used in ML-BENCH-L, we emphasize the agent's effectiveness in fulfilling user requirements through interactive execution rather than predetermined outputs (**Success Rate**). Success is defined by agents correctly following repository-documented workflows, matching expected execution patterns, and producing outputs in the required format. Unlike the stochastic nature of ML tasks, which complicates direct output validation, our deterministic evaluation framework focuses on reproducible and consistent criteria. These include environment setup, dependency management, and correct API usage, verified by human annotators with high agreement (0.92 Cohen's kappa). This methodology ensures reliability by avoiding stochastic variations and emphasizing correct repository interaction over final model performance.

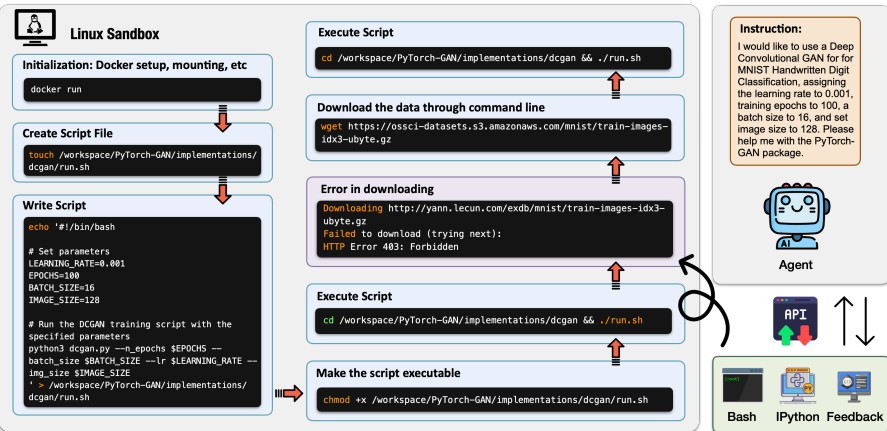

Figure 5: An example of agent execution logs demonstrating the interactive nature of ML-BENCH-A.

## 4.2 EXPERIMENTAL RESULTS

In Table 4, we detail the performance of various agents such as AutoGen, SWE-Agent, and Aider, as well as OpenDevin equipped with diverse GPT language models, evaluated on a quarter subset of the test set. OpenDevin, utilizing GPT-4o (model name: `gpt-4o-2024-05-13`), achieved the best

results, striking an excellent balance between cost and performance. The success rate, the number of instances successfully solved, and the average cost per solved instance were the critical metrics for this evaluation. As demonstrated by the varied performance of agents using the identical `gpt-4-1106` model, it becomes evident that the choice of agent framework significantly impacts the effectiveness of an agent. This discrepancy in success rates and average costs accentuates the potential for future advancements in agent architecture to enhance performance further.

## 5 ANALYSIS

### 5.1 DATA LEAKAGE

Since the repositories we selected are quite popular and likely to have appeared in the model's pre-training data, we found that sometimes even when the model is provided with Bash script information instead of Python code, it still tends to generate code snippets that closely resemble those in the original data. We believe that in this scenario, data leakage has affected the model's ability to follow instructions. To mitigate the impact of data leakage, we verify that the type and parameters of the generated results align with user instructions before execution. We show the updating status for all repositories in Appendix G.1.

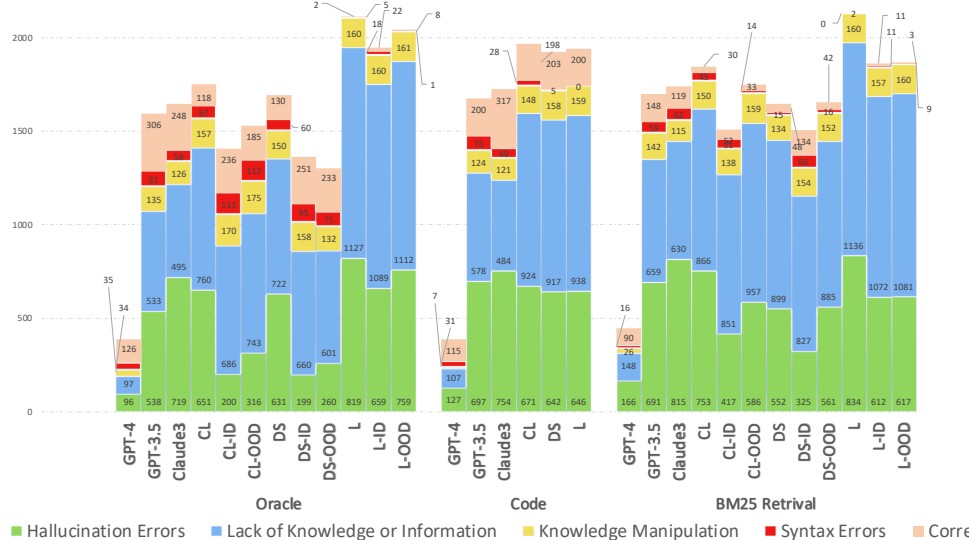

Figure 6: Quantification of models and settings errors with five attempts. The total statistic results are 1,300 for the full test set. Statistical results that exceed these numbers are caused by multiple errors made on one result simultaneously. For models, CL stands for CodeLlama, DS stands for deepseek-coder, and L stands for Llama-2. Raw means that the model is not fine-tuned. ID means that the model is fine-tuned in an in-distribution setting. OOD means that the models are fine-tuned in an out-of-distribution setting. Here, Claude3 stands for `Claude-3-Haiku`.

### 5.2 ERROR ANALYSIS

By analyzing the execution log, we find that the errors of models and agents fall into five categories:

**Hallucination Errors (E1)**: These errors include instances when the models misinterpreted the user's intention, misplaced Python code and bash script, or generated random or irrelevant code.

**Lack of Knowledge or Information (E2)**: This type of error primarily stems from the model's inability to fulfill user requirements based on crucial parameters. Possible types of errors are as follows:

1. Code inner information. The models sometimes lack sufficient information necessary to satisfy the user's requirements. For instance, this deficiency might involve missing parameter names (`--lr` and `--learning-rate`) or unavailable options (it only accepts 'Citeseer' when the input given was 'citeseer').

2. Domain knowledge. The models sometimes lack the domain-specific knowledge required to handle certain instances. For example, in BERT, the models simultaneously generated `--case=True` and `--model=uncased`.

3. Grammar knowledge. This happens when the models incorrectly identify and handle certain command line symbols. Like the $ symbol, which could affect execution.

4. Local data information. The models were not supplied with enough crucial parameter information for specific commands, leading to the inability to find paths and successful execution. While less common, this error was particularly notable in the OpenCLIP.

**Knowledge Manipulation (E3)**: Take BERT, for instance, where the model needed to integrate `DIR=/model/` and `--model_dir=$DIR` to form `--model_dir=/model`. There were also cases where it couldn't merge `/model_name` in `/data` into a proper path format like `/data/model_name`. Similar incidents were observed in OpenCLIP.

**Syntax Errors (E4)**: These errors cover instances of incorrect code generation of syntax errors instead of hallucination, mainly Python syntax errors such as the use of undefined variables. These errors arise from cases that prefer generating Code.

**Operational Error (E5 – ML-BENCH-A Only)**: These errors only occur in ML-BENCH-A. Less powerful agents, such as those using base models like GPT-3.5, may have difficulty understanding the task in the execution sandbox. In these cases, the agent gives up easily or exits without finishing the tasks. Interestingly, these errors are less observed in advanced model settings.

On ML-BENCH-L, we find that Claude 3 has more hallucinations than GPTs for closed-source models. However, its ability to fulfill requirements (represented by E2) is better than that of GPT-3.5. Under the retrieval setting, neither GPT-3.5 nor Claude 3 exhibits an increase in hallucination but an increase in the proportion of E2 compared to the Code setting. For GPT-4, both E1 and E2 increased because GPT-4 generates code without considering the content in cases involving task-irrelevant information, except for GPT-3.5 and Claude 3.

Compared to the Code setting, Oracle provision decreases the quantities of E1 and E2, while the differences in E3 and E4 are insignificant. This suggests that whether an Oracle is provided does not significantly affect the closed-source models' knowledge manipulation and Python code generation abilities. We tend to attribute these to the models' inherent ability rather than the reference.

The ML-BENCH-A showcases potential in incorporating feedback from the experience, leading to fewer E1 and E4 errors. Yet, compared to ML-BENCH-L, ML-BENCH-A is more prone to E2 and E3 errors due to file type discrepancies from the set output type. Because of the sandbox environment's great flexibility, we observe increased hallucinations with each step, including installing conflicting dependencies and navigating incorrect or even illegal directories. Meanwhile, E5 happens in less powerful agents. The agent often ignores the instruction to export the final solution to `run.sh`, or refuses to answer the question immediately after entering the virtual sandbox. Compared to GPT-4, GPT-4o shows enhanced abilities in sourcing relevant information from repositories, leading to markedly fewer errors associated with E2 and E3. Yet, it exhibits a tendency toward more frequent hallucinations than GPT-4, for instance, installing incorrect packages. For a detailed error analysis and quantitative performance visualization on both setups, refer to Appendix G.6 and G.7, and for examples running on both setups, refer to Appendix G.8 and G.9.

## 6 CONCLUSION

ML-Bench addresses the limitations of existing benchmarks in comprehensively evaluating model performance across real-world ML workflows. It simulates the complete ML development process, from environment configuration to code execution. Our tasks require models to retrieve supporting evidence, generate code, and set hyperparameters correctly, as well as download/install existing datasets, models, & packages. We introduce ML-BENCH-L and ML-BENCH-A, two distinct evaluation setups assessing LLMs' code generation capabilities and agents' end-to-end task execution abilities, respectively. Results show GPT-4 achieving a Pass@5 rate over 50% in ML-BENCH-L and a 76.47% success rate in ML-BENCH-A, highlighting areas for improvement in handling hallucinations and bash script generation.

LIMITATION

Our study, while comprehensive within its scope, is subject to certain limitations that stem primarily from linguistic and data source constraints.

**Models Limitation**    We acknowledge that the scope of our benchmark might not entirely encapsulate the breadth of available open-source models. While we have conducted extensive tests using a variety of models beyond the results presented in the paper, including but not limited to:

- `mistralai/Mistral-7B-Instruct-v0.3`
- `mistralai/Mixtral-8x22B-Instruct-v0.1`
- `Qwen/Qwen1.5-72B-Chat`
- `Qwen/Qwen1.5-110B-Chat`
- `Qwen/Qwen2-72B-Instruct`
- `codellama/CodeLlama-34b-Instruct-hf`
- `meta-llama/Meta-Llama-3.1-8B-Instruct-Turbo`
- `meta-llama/Meta-Llama-3.1-70B-Instruct-Turbo`
- `meta-llama/Meta-Llama-3.1-405B-Instruct-Turbo`

Due to space constraints, detailed results from these models were not included in the manuscript. We emphasize that the primary objective of our benchmark is not to be an exhaustive repository of the latest open-source models but rather to establish a robust and versatile benchmark framework. Our goal is to inspire the community to develop better ML agents.

Our benchmark, ML-Bench, is designed to be widely applicable and has already seen extensive adoption within the community. By providing a comprehensive and practical evaluation framework, we aim to pave the way for future advancements in the development of ML agents, regardless of the specific models used.

**Data Source Limitation - Reliance on GitHub Repositories in English**    Our reliance on GitHub repositories with documents exclusively in English introduces a selection bias. GitHub, while rich in open-source projects and documentation, may not comprehensively represent the broader landscape of software development practices and trends globally. This choice potentially overlooks significant contributions and insights from non-English-speaking communities. This limitation might impact the development of tools and models tailored to a more diverse set of programming environments and community needs.

**Methodological Limitation - Relying on Pre-built Machine Learning Packages**    Our methodology utilized existing machine learning packages instead of developing algorithms from scratch. While this approach allowed us to leverage well-established, tested, and optimized tools, it also introduces certain constraints. Dependence on pre-built packages means our work is confined to the capabilities and limitations of these tools. This reliance could limit our ability to fully explore novel or unconventional approaches possible with custom-built algorithms. Moreover, this choice potentially impacts the reproducibility and customization of our findings. Researchers who seek to build upon our work may encounter similar constraints imposed by the pre-built packages we utilize. These limitations can hinder innovation and adaptation in different contexts or for specific usage.

**Scope Limitation - Tasks Limited to README File Descriptions**    By strictly adhering to the specified tasks, our study may overlook potential applications or challenges not explicitly documented in README. This limitation can result in a narrower understanding of the tools we examined, as it fails to explore their full potential and applicability. The reliance on README descriptions also assumes that these documents comprehensively and accurately reflect all relevant aspects of the repositories, which may not always be accurate. Important tasks or nuances might be undocumented or underrepresented in these files.

## ETHICS STATEMENT

In our work, we have carefully considered the ethical implications of our work, particularly in data annotation and related activities. Our methodologies and processes have been meticulously designed to ensure they are free from moral concerns. We affirm that our research practices, including data handling, have been conducted with the utmost integrity and in compliance with ethical standards.

Our approach has been guided by principles prioritizing respect for data integrity, transparency in our methods, and adherence to established ethical guidelines.

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

## A REGARDING BASH SCRIPT GENERATION

We would like to clarify that ML-Bench encompasses a much broader range of tasks than just bash script generation. Specifically:

**ML-Bench-L**: This benchmark component includes tasks that require generating both bash scripts and Python code. The diversity in task types reflects the varied nature of ML workflows, where both scripting and programming play essential roles.

**ML-Bench-A**: In this more complex setup, agents are required to interact with the environment using a combination of bash commands and Python code with tools such as Jupyter Notebooks. This approach closely mimics ML practitioners' actual workflow, who often switch between command-line operations and code execution in interactive environments.

Including bash script tasks, alongside Python code generation and execution, is intentional and reflects the reality of ML development workflows. Many real-world ML tasks involve a combination of environment setup (often done via bash commands), data preprocessing, and model implementation (typically done in Python).

- (1) ML-Bench is not limited to bash script tasks. Our benchmark includes many task types, encompassing both bash script and Python code generation.
- (2) The tasks in ML-Bench are carefully designed to mirror the authentic workflows of ML practitioners. This approach has been recognized as meaningful and valuable in previous and follow-up works.

In conclusion, while bash script tasks are indeed part of ML-Bench, they represent only one component of a much broader and more complex set of challenges. Our benchmark's strength lies in its comprehensive coverage of the ML development lifecycle, addressing meaningful scenarios highly relevant to real-world ML practice and research. We believe this approach provides valuable insights into model capabilities that complement existing benchmarks.

## B   ML-BENCH'S SCOPE AND SIGNIFICANCE

While SWE-Bench focuses on resolving GitHub issues, ML-Bench addresses a distinct yet equally critical aspect of real-world software development: the ability to utilize the existing code in machine-learning contexts. This workflow closely mirrors common scenarios faced by ML engineers (like the difference between MLE and SWE).

ML-Bench evaluates several crucial capabilities that are not explicitly tested in SWE-Bench:

- a. Many tasks involve setting appropriate hyperparameters or configuration options, requiring an understanding of both the code and the underlying machine learning concepts.
- b. ML-Bench includes 18 diverse repositories (compared to SWE-Bench's 12), challenging models to adapt to various ML data types (as illustrated in Figure 2 of our paper).
- c. ML-Bench requires a combination of code comprehension, environment setup, and execution that closely mirrors the day-to-day activities of ML practitioners. We include tasks such as environment setup, dependency management, and data downloading – all crucial skills in practical ML development.

We added two examples that illustrate these tasks' complexity and real-world relevance. For instance:

**Example 1: Understanding and Implementing Complex Neural Architectures.**

While our dataset doesn't typically include tasks requiring the creation of training loops from scratch, it does involve understanding and correctly utilizing complex model architectures. For example:

```
1  {"github_id": 9, "github": "https://github.com/xmu-xiaoma666/External-
      Attention-pytorch",
2  "repo_id": 26, "path": "./",
3  "arguments": "{'data': '(50,49,512)', 'model': 'ExternalAttention', '
      argument3': 'torch'}",
4  "instruction": "I am in possession of a data input in the shape of
      (50,49,512).
5  My intention is to deploy this within the realm of External Attention
      Usage.
6  My aim is to successfully complete Attention layers using the fighting cv
      library.
7  However, after accomplishing this, I would also want to know the output
      shape.
8  May you kindly assist in crafting the necessary coding elements?",
9  "oracle": " from model.attention.ExternalAttention import
      ExternalAttention\n
10 import torch\n\ninput=torch.randn(50,49,512)\nea = ExternalAttention(
      d_model=512,S=8)\noutput=ea(input)\nprint(output.shape)\n\n",
11 "type": "Python Code", "id": 268,
12 "prefix_code": "git clone https://github.com/xmu-xiaoma666/External-
      Attention-pytorch.git\n
13 cd External-Attention-pytorch"}
```

This example demonstrates that the model needs to:

- Understand the structure of the External Attention model.
- Correctly import and initialize the model with appropriate parameters.
- Generate input data of the correct shape.
- Apply the model to the input data.
- Print the output shape.

**Example 2: Parameter Understanding and Customization.**

Our tasks often require models to understand and correctly use parameters defined across multiple files. The expected output for the above task showcases this:

```python
import torch
from model.attention.ExternalAttention import ExternalAttention
input_size = (50, 49, 512)
parser = argparse.ArgumentParser()
parser.add_argument("--d_model", type=int, default=512, help="
    dimensionality of the model")
parser.add_argument("--S", type=int, default=8, help="number of attention
    heads")
opt = parser.parse_args()
input = torch.randn(*input_size)
ea = ExternalAttention(d_model=opt.d_model, S=opt.S)
output = ea(input)
print(output.shape)
```

This solution demonstrates that the model needs to:

- Understand the purpose and usage of argparse for parameter customization.
- Correctly set up default values and help messages for each parameter.
- Use these parameters when initializing the ExternalAttention model.

## B.1 SWE-BENCH AND ML-BENCH

While SWE-Bench indeed simulates the real-world scenario of resolving GitHub issues, which often involves modifying specific functions or files within a repo and is crucial for software development, ML-Bench aims to capture another class of tasks frequently encountered by machine learning (ML) practitioners - how to train and deploy models on specific datasets.

For example, consider a scenario where a data scientist needs to train a sentiment classification model on a specific dataset using a particular ML library. This task involves:

- Understanding the structure of the ML library repo
- Setting up the appropriate environment
- Preprocessing the dataset
- Selecting and implementing a suitable model architecture
- Training the model with appropriate hyperparameters
- Evaluating the model's performance

This end-to-end ML workflow is not typically covered by SWE-Bench but is a core focus of ML-Bench.

We view the two benchmarks as complementary, jointly advancing research on LLMs' applications in software development and ML.

| Criteria | SWE-Bench | ML-Bench (Ours) |
|---|---|---|
| Repo. Understanding | ✓ | ✓ |
| README. Understanding | ✗ | ✓ |
| Cross-File Retrieval | ✓ | ✓ |
| Package Installation | ✗ | ✓ |
| Data Downloading | ✗ | ✓ |
| ML Model Training | ✗ | ✓ |
| Environment Configuration | ✗ | ✓ |
| Evaluation | Success Rate | Pass@K / Success Rate |
| # of Repositories | 12 | 18 |
| # of Tasks | 2,300 | 9,641 |
| Focus Area | General SWE | ML-specific |

Table 5: Comparison of SWE-Bench and ML-Bench.

Key distinctions between ML-Bench and SWE-Bench include:

**a) Range of Tasks:**

- **ML-Bench:** Includes environment configuration, dataset and model downloading, code generation, and execution testing.
- **SWE-Bench:** Primarily bases its evaluation on the correctness of problem-solving for individual issues.

This comprehensive approach in ML-Bench more closely mimics the end-to-end workflow of ML practitioners.

**b) Repository Understanding:**

- **ML-Bench:** Requires models to retrieve relevant code snippets from the repository as references for writing scripts that invoke repository functions or workflows.
- **SWE-Bench:** Focuses on searching the repository to locate and modify specific code segments to resolve predefined issues.

**c) Documentation Utilization:**

- **ML-Bench:** Explicitly evaluates the model's ability to understand and utilize documentation such as README files.
- **SWE-Bench:** This aspect is not explicitly addressed.

**d) Package Installation and Data Downloading:**

- **ML-Bench:** Includes tasks related to package installation and data downloading, which are critical steps in ML workflows.
- **SWE-Bench:** These tasks are not covered.

# C    QUANTIFYING THE CONTEXT REQUIRED

To quantify the context needed for each instance in the benchmark, we randomly sampled 20 examples and performed a detailed analysis.

We have added a table 6 quantifying the context required for each task type, including statistics on the average amount of code that needs to be understood, the distribution of relevant information across different files, and the importance of README files versus actual code.

| Metric | Value |
|---|---|
| Average number of relevant files | 3.6 |
| Average lines of code in relevant files | 414 |
| Percentage of tasks requiring README understanding | 85% |
| Percentage of tasks requiring code understanding | 95% |
| Average depth of relevant code in repository (line number) | 27,524 |
| Percentage of tasks requiring understanding of multiple files | 75% |
| Average number of functions/classes to be understood per task | 3.8 |
| Percentage of tasks requiring understanding of dependencies | 70% |
| Percentage of tasks requiring understanding of data structures | 65% |
| Percentage of tasks involving API usage | 80% |

Table 6: Metrics related to different aspects of the tasks.

This analysis shows that while README files are important, understanding the code itself is crucial for many tasks. Contrary to the reviewer's impression, our benchmark heavily emphasizes code understanding. 100% of tasks require comprehension of actual code, not just README files. While 85% of tasks do involve README files, this is in addition to, not instead of, code understanding. READMEs often provide crucial context for code interpretation.

Additionally, we have added a table below quantifying the required context for each task type, including an average number of relevant files, lines of code, and key information distribution across repository components.

- Required context length: Number of tokens in relevant files
- Average number of relevant files: Mean number of files pertinent to the task
- Relevant lines of code: Number of lines in relevant files
- Average depth of code in repository (line): Line number where relevant files first appear

| Repository name | Required context length | Average number of relevant files | Relevant lines of code | Average depth of code in repository (line) |
|---|---|---|---|---|
| dgl | 974 | 6.2 | 905 | 220624 |
| bert | 3204 | 1 | 494 | 6738 |
| Pytorch-GAN | 988 | 1 | 168 | 5004 |
| Vid2Vid | 1221 | 6 | 892 | 4042 |
| Time-SL | 1753 | 6 | 936 | 341 |
| Py-IM | 8712 | 8 | 1108 | 3255 |
| Learning 3D | 1330 | 1 | 223 | 7201 |
| Music | 1851 | 3.4 | 461 | 14012 |
| External-AP | 107 | 1 | 24 | 16425 |
| Open-CLIP | 882 | 1.2 | 267 | 14087 |
| IF | 1514 | 4 | 360 | 3410 |
| Segment Anything | 1423 | 5 | 678 | 9572 |
| ESM | 811 | 3 | 339 | 5516 |
| LAVIS | 1093 | 3.1 | 346 | 2640 |

# D   COST OF HUMAN ANNOTATION

Eight computer science graduate students with proficient programming abilities contributed to the data annotation.

Each repository took approximately 5-10 hours to annotate.

Each student spent about 30 hours constructing data, crafting prompts, reviewing code for retrieval, and performing quality control checks.

While human involvement is required, we believe the scale of the effort is manageable and comparable to other code-based benchmarks. The total time invested (approximately 240 hours) is significantly less than some other benchmarks. For example, the DS1000 paper reported that five authors spent 1200 hours on data construction.

In addition, all code-based benchmarks, including DS1000, SWE-Bench, and RepoBench, require human annotation for data construction. Our approach is significantly less time-intensive than these established benchmarks.

Furthermore, we have implemented semi-automated processes for certain aspects of our benchmark creation, as described in Figure 3 of our paper. README selection and Instruction Generation can be partially automated using GPT, reducing the manual workload.

Lastly, while the annotation process does require significant effort, we believe the resulting benchmark provides unique and valuable insights into model performance on real-world ML tasks. The depth and complexity of our tasks justify the investment in human annotation.

Importantly, we focus on providing a high-quality dataset with a reasonable yet manageable size that allows reliable assessment of LLMs' capabilities in this task, rather than just curating a large-scale evaluation dataset. This approach ensures that each task in ML-Bench is carefully curated and validated, maintaining a high standard of quality and relevance to real-world ML workflows. By prioritizing quality over quantity, we aim to offer a more nuanced and accurate evaluation of LLM performance in the context of machine learning development tasks.

BENCHMARK CREATION PROCESS

Our benchmark creation process incorporates semi-automated elements. Specifically:

**a) README Selection:**

- Partially automated using LLMs to identify relevant sections.

**b) Instruction Generation:**

- Leverages GPT models to draft initial task descriptions, which human annotators then refine.

These steps significantly reduce the manual workload while maintaining task quality.

Our approach is comparably efficient to other established code-based benchmarks. For instance, our total annotation time (approximately 240 hours) is significantly less than the 1200 hours reported for DS-1000 construction.

IMPORTANCE OF HUMAN ANNOTATION

Human annotation is crucial for ensuring ML-Bench's task quality and authenticity, particularly given the unique challenges of machine learning repositories. Here's why:

Machine learning repositories often contain specialized knowledge and complex workflows that require deep understanding. Our human annotators, computer science graduate students with proficient programming abilities, bring essential domain expertise to the task-creation process. This expertise is critical for crafting tasks that accurately reflect real-world ML development challenges.

We initially explored using GPT to generate tasks automatically, but the results were unsatisfactory, leading us to abandon this approach. This experience highlights the high quality of our manually curated dataset.

The necessity for human annotation in ML-Bench aligns with current practices in code benchmarks; to our knowledge, all existing code benchmarks (e.g., SWE-bench, RepoBench, DS-1000) require manual annotation and cannot be easily scaled through automation.

ML-Bench tasks, especially in the ML-Bench-A setup, often involve multi-step processes, including environment setup, repository navigation, and ML-specific tasks. Human annotators can craft cohesive, end-to-end scenarios that authentically represent these complex workflows, which would be challenging to generate automatically.

# E    DEFINITION OF OUR TASKS

We refine our task definition to ensure clarity. Here's a concise definition, along with an example.

We've elaborated on our two distinct setups:

**ML-Bench-L:** Evaluates models' capacity to complete tasks within a pre-configured deployment environment, translating text instructions to simple bash or Python code with clearly defined parameters. The environment is already set up with the necessary dependencies and datasets.

**ML-Bench-A:** Introduces a secure Linux sandbox environment where agents start with an empty Docker container and must iteratively execute commands and code blocks to set up the environment, install dependencies, download datasets, and finally execute the task, emulating the full workflow of a human coder.

| Setup | Task Definition |
|---|---|
| ML-Bench-L | Given a GitHub repository, all its files, an instruction, and arguments, generate executable Bash or Python code that utilizes functions or models from the repository in line with the user instruction and arguments. The execution environment is pre-configured. |

| ML-Bench-A | Given access to an empty Docker environment, a GitHub repository URL, an instruction, and arguments $\mathcal{A}$, the task is to iteratively create a Docker container setup and generate executable Bash or Python code to utilize repository functions/models to fulfill the given instruction and arguments. |
|---|---|

**Examples**

| Setup | Example |
|---|---|
| ML-Bench-L | Repository: Image generation model repository
Instruction: Generate an image based on a text description
Arguments: prompt="a girl riding a horse", ckpt="SD2_1_v_model.ckpt"
Expected Output: `python txt2img.py -prompt "a girl riding a horse" -ckpt SD2_1_v_model.ckpt` |
| ML-Bench-A | Repository URL: `https://github.com/example/image-gen-repo.git`
Instruction: Generate an image based on a text description
Arguments: prompt="a girl riding a horse", ckpt="SD2_1_v_model.ckpt"
Expected Actions and Output:
1. `git clone https://github.com/example/image-gen-repo.git`
2. `cd image-gen-repo`
3. `pip install -r requirements.txt`
4. `wget https://example.com/models/SD2_1_v_model.ckpt`
5. `python txt2img.py -prompt "a girl riding a horse" -ckpt SD2_1_v_model.ckpt` |

We also expand our description of the tasks and provide a more detailed breakdown for both ML-Bench-L and ML-Bench-A.

ML-Bench-L tasks are primarily focused on code generation and can be categorized into two types based on the language:

**a) Bash Script Generation:**

- Average length: 176 characters
- Example: Setting up environment variables, running specific scripts

**b) Python Code Generation:**

- Average length: 244 characters
- Example: Implementing data preprocessing, model initialization, or inference steps

For ML-Bench-A, we conducted a detailed analysis of 20 randomly sampled examples. Our analysis revealed that each example/instance in ML-Bench-A comprehensively evaluates the agent's capabilities across three key areas:

**a) Environment Setup:**

- Package installation
- Dataset downloading
- Model downloading

**b) Repository Navigation and Understanding:**

- Identifying relevant files and functions
- Understanding dependencies between different components

**c) ML-Specific Tasks:**

- Model initialization and configuration
- Data preprocessing
- Model training (in select cases)
- Model inference

It's important to note that unlike ML-Bench-L, where tasks are categorized by language, each ML-Bench-A example requires the agent to perform actions across all three categories. This design ensures a comprehensive evaluation of the agent's ability to handle complex, multi-step ML workflows.

We have also added information on the complexity of tasks, including:

- Average number of relevant files per task: 3.5
- Average lines of code in relevant files: 514
- Percentage of tasks requiring multi-file understanding: 70%

EXAMPLE 1: ML-BENCH-L (PYTHON CODE GENERATION)

**Instruction:** Generate code to perform inference using the BERT model on the input text "Hello, world!"
**Repository:** bert
**Expected Output:**

```python
import torch
from transformers import BertTokenizer, BertForSequenceClassification

tokenizer = BertTokenizer.from_pretrained('bert-base-uncased')
model = BertForSequenceClassification.from_pretrained('bert-base-uncased'
    )

input_text = "Hello, world!"
inputs = tokenizer(input_text, return_tensors="pt")
outputs = model(**inputs)

print(outputs.logits)
```

EXAMPLE 2: ML-BENCH-A (COMPLETE WORKFLOW)

**Instruction:** Set up the Stable Diffusion environment, download the model, and generate an image of a "cat on a beach"
**Repository:** Stable Diffusion
**Expected Actions:**

**1. Environment Setup:**

```
git clone https://github.com/CompVis/stable-diffusion.git
cd stable-diffusion
pip install -r requirements.txt
```

**2. Model Download:**

```
wget https://github.com/CompVis/stable-diffusion/releases/download/v1.4/
    sd-v1-4.ckpt
```

**3. Repository Navigation:** Identify the main script for image generation (`scripts/txt2img.py`)

**4. ML-Specific Task (Image Generation):**

```
python scripts/txt2img.py --prompt "cat on a beach" --plms
```

# F  MODEL DOWNLOADING

In our ML-Bench-A, the agent is responsible for locating and downloading the correct trained model. This process is part of the task and is not pre-configured.

The agent may download the correct trained DL model or an incorrect one, which mimics real-world scenarios where developers might encounter issues with model compatibility or versioning.

Our testing environment executes the code generated by the agent, including any model loading steps.

The success rate is measured based on successful model loading, correct inference, and appropriate output formatting. If the agent downloads an incorrect model or fails to set up the environment correctly, it will not pass the test, reflecting real-world challenges in ML workflows.

This approach ensures that our benchmark evaluates not just code generation, but also the agent's ability to understand and correctly set up the entire ML pipeline, including model selection and environment configuration.

# G  DATASET DETAILS

## G.1  DETAILS OF SELECTED GITHUB REPOSITORIES

As depicted in Table 10, our selection encompasses a range of GitHub repositories varying from language and graph models to multimodal and time-series models. Each repository is chosen for its high-quality contributions to the field and its popularity among the development community, indicated by the number of stars. The repositories, diverse in their updates and number of README files, provide a snapshot of the current landscape of models available on GitHub.

Table 10: Comprehensive information on selected GitHub repositories. The column labeled "#README" refers to the number of README files contained within each listed GitHub repository.

| Domain | GitHub | Stars | URL | #README | Last Updated |
|---|---|---|---|---|---|
| Language Model | BERT | 35,693 | https://github.com/google-research/bert | 1 | 2020.03.11 |
| | Tensor2Tensor | 14,280 | https://github.com/tensorflow/tensor2tensor | 9 | 2023.04.01 |
| Graph Model | DGL | 12,429 | https://github.com/dmlc/dgl | 154 | 2023.11.16 |
| Biomedical Model | ESM | 2,462 | https://github.com/facebookresearch/esm | 8 | 2023.06.27 |
| | MedicalZooPytorch | 1,516 | https://github.com/black0017/MedicalZooPytorch | 21 | 2022.02.07 |
| Vision Model | PyTorch-GAN | 14,947 | https://github.com/eriklindernoren/PyTorch-GAN | 1 | 2021.01.07 |
| | Learning3d | 579 | https://github.com/vinits5/learning3d | 1 | 2023.10.24 |
| | External-Attention-pytorch | 9,949 | https://github.com/xmu-xiaoma666/External-Attention-pytorch | 1 | 2023.10.25 |
| | Pytorch-image-models | 30,400 | https://github.com/huggingface/pytorch-image-models | 1 | 2023.11.09 |
| Audio Model | Muzic | 3,866 | https://github.com/microsoft/muzic | 8 | 2023.12.06 |
| Multi-Modality | LAVIS | 7,300 | https://github.com/salesforce/lavis | 8 | 2023.09.25 |
| | IF | 7,237 | https://github.com/deep-floyd/if | 1 | 2023.06.03 |
| | OPEN-CLIP | 6856 | https://github.com/mlfoundations/open_clip | 1 | 2023.11.01 |
| | Stable Diffusion | 31,506 | https://github.com/Stability-AI/stablediffusion | 1 | 2023.03.25 |
| | Segment-Anything | 11,976 | https://github.com/IDEA-Research/Grounded-Segment-Anything | 3 | 2023.12.11 |
| Video | Vid2Vid | 8,393 | https://github.com/NVIDIA/vid2vid | 2 | 2019.07.04 |
| Time-Series Model | Time-Series-Library | 2,670 | https://github.com/thuml/Time-Series-Library | 1 | 2023.11.10 |

## G.2  TEMPLATES FOR DIVERSE INSTRUCTION CONSTRUCTION

To generate precise instructions, we have developed a range of instruction templates that serve as the foundation for the tasks, ensuring that all necessary arguments are present and accounted for. For a detailed look, please refer to the instruction templates outlined in Table 11. These templates are instrumental in guiding the generation process, embodying diversity and adaptability while maintaining the integrity of the input information. The templates in the table guide users to regenerate similar instructions with varied expressions while keeping the task requirements intact, promoting diversity in instructional outputs for the model to process and execute.

Table 11: Examples of constructed instructions for diverse task templates. In this context, "ESMFold" refers to a task related to protein structure prediction using a particular machine learning model or function.

---

**Task Description**

You need to use your creativity and rewrite each query three times. Make sure that all the required arguments are provided. The instructions should be diverse, and you are encouraged to write long instructions. The instructions should be very different.

**Example 1: ESMFold Structure Prediction**

**Input:** I have a data sequence MKTVRQERLKSI, and I want to use `esm1v_t33_650M_UR90S_1()` to finish the ESMFold Structure Prediction task ...

**Required Arguments:**

- **Sequence:** MKTVRQERLKSI

- **Model:** `esm1v_t33_650M_UR90S_1()`

- **Task:** ESMFold Structure Prediction

- ...

**Output:**

1. **Instruction:** Input data: MKTVRQERLKSI, Model: `esm1v_t33_650M_UR90S_1()` ... Can you please design the model for me?

2. **Instruction:** I possess a data sequence MKTVRQERLKSI and I'd like to use `esm1v_t33_650M_UR90S_1()` for the ESMFold Structure Prediction task ...

3. **Instruction:** I have an interesting sequence MKTVRQERLKSI and I'm planning to utilize `esm1v_t33_650M_UR90S_1()` for ESMFold Structure Prediction ...

**Example 2: Another ESMFold Structure Prediction Task**

**Input:** I want to use `esmfold_v1()` to finish the ESMFold Structure Prediction task ...

**Required Arguments:**

- **Model:** `esmfold_v1()`

- **Task:** ESMFold Structure Prediction

- **Learning Rate:** 0.04

- ...

**Output:**

1. **Instruction:** Do you know ESM? I need to perform the ESMFold Structure Prediction task using `esmfold_v1()`. I'd like to set the learning rate to 0.04 ...

2. **Instruction:** I'm working on ESMFold Structure Prediction and plan to use `esmfold_v1()` with a learning rate of 0.04 ...

3. **Instruction:** I'm looking to employ `esmfold_v1()` for the ESMFold Structure Prediction task, setting the learning rate to 0.04 ...

**Instruction: Construct the Instructions for SQUAD2.0 Fine-tuning Task**

You must follow the above examples, keep the information unchanged, and write some new instructions. Your output should be diverse. You are encouraged to write long instructions. The instructions should be very different.

**Input:** I'm interested in utilizing `./model/cased_L-12_H-768_A-12` for fine-tuning the SQUAD2.0 task. I plan to establish a train batch size of 128. Could you assist me in writing the necessary code?

**Required Arguments:**

- **Model Path:** `./model/cased_L-12_H-768_A-12`

- **Task:** SQUAD2.0

- ...

**Output:**

1. **Instruction:** [Your Instruction Here]

2. **Instruction:** [Your Instruction Here]

3. **Instruction:** [Your Instruction Here]

---

### G.3 TOKEN LENGTH ANALYSIS

For a detailed examination of the token lengths encountered during our model evaluations, we present two tables (Table 12 and 13) comparing the token counts across a variety of experimental settings. The measurements were made using the CodeLlama tokenizer and GPT-3.5 tokenizer.

Table 12: Average input token length of instructions for CodeLlama and GPT-3.5* in various experimental settings. *We use tiktoken (https://github.com/openai/tiktoken) to calculate the token number for GPT-3.5.

| Repository | CodeLlama | | | | | | GPT 3.5 | | | | | |
|---|---|---|---|---|---|---|---|---|---|---|---|---|
| | Code | | Retrieval | | Oracle | | Code | | Retrieval | | Oracle | |
| | Train | Test | Train | Test | Train | Test | Train | Test | Train | Test | Train | Test |
| **In-Distribution (ID)** | | | | | | | | | | | | |
| DGL | 5,466,687 | 5,466,687 | 312 | 2,603 | 179 | 138 | 4,455,349 | 4,455,349 | 275 | 2,011 | 143 | 110 |
| BERT | 138,445 | 138,445 | 401 | 344 | 372 | 375 | 112,104 | 112,104 | 335 | 280 | 287 | 290 |
| ESM | 27,107,031 | 27,107,031 | 585 | 438 | 177 | 173 | 22,227,765 | 22,227,765 | 486 | 273 | 139 | 136 |
| Py-GAN | 146,570 | 146,570 | 532 | 897 | 314 | 314 | 119,454 | 119,454 | 433 | 744 | 268 | 268 |
| Lavis | 16,827,026 | 16,827,026 | 471 | 401 | 1984 | 1984 | 13,714,026 | 13,714,026 | 372 | 325 | 1547 | 1547 |
| External-Attention (EAP) | 449,381 | 449,381 | 1155 | 526 | 105 | 118 | 346,898 | 346,898 | 857 | 412 | 69 | 80 |
| If | 68,316 | 68,316 | 1390 | 1,642 | 3023 | 3023 | 55,677 | 55,677 | 1119 | 1,330 | 2367 | 2367 |
| vid2vid | 146,696 | 146,696 | 408 | 1615 | 556 | 565 | 111,783 | 111,783 | 338 | 481 | 416 | 416 |
| OpenCLIP | 6,143,829 | 6,143,829 | 415 | 491 | 5420 | 5420 | 5,037,939 | 5,037,939 | 350 | 405 | 4397 | 4397 |
| TSL | 337,114 | 337,114 | 382 | 902 | 345 | 345 | 273,062 | 273,062 | 315 | 731 | 276 | 276 |
| **Out-Of-Distribution (OOD)** | | | | | | | | | | | | |
| Grounded-SAM | / | 16,726,416 | / | 898 | / | 164 | / | 13,715,662 | / | 754 | / | 113 |
| Py-IM | / | 5,608,249 | / | 8,025 | / | 89 | / | 4,542,681 | / | 6,415 | / | 68 |
| muzic | / | 13,325,828 | / | 616 | / | 83 | / | 10,860,549 | / | 507 | / | 64 |
| Learning3D | / | 320,157 | / | 640 | / | 50 | / | 256,110 | / | 596 | / | 45 |
| SD | 258,096 | / | 501 | / | 234 | / | 209,058 | / | 412 | / | 183 | / |
| MedZooPy | 2,701,443 | / | 1,302 | / | 133 | / | 2,150,168 | / | 1,101 | / | 99 | / |
| TCL | 18,696,614 | / | 345 | / | 116 | / | 15,114,250 | / | 291 | / | 96 | / |
| Tensor2Tensor | 4,598,727 | / | 501 | / | 192 | / | 3,678,980 | / | 432 | / | 153 | / |

Table 13: Average output token length of code for GPT-3.5* and CodeLlama to generate across different datasets (Train Set, Test Set, 1/4 Test Set) for various repositories, separated by Python Code and Bash Script. *We use tiktoken (https://github.com/openai/tiktoken) to calculate the token number for GPT-3.5.

| Repository | Train Set | | | | Test Set | | | | 1/4 Test Set | | | |
|---|---|---|---|---|---|---|---|---|---|---|---|---|
| | GPT-3.5 | | CodeLlama | | GPT-3.5 | | CodeLlama | | GPT-3.5 | | CodeLlama | |
| | Python | Bash | Python | Bash | Python | Bash | Python | Bash | Python | Bash | Python | Bash |
| **In-Distribution (ID)** | | | | | | | | | | | | |
| DGL | / | 21.15 | / | 28.05 | / | 18.24 | / | 24.33 | / | 21.60 | / | 28.40 |
| BERT | / | 121.98 | / | 181.60 | / | 120.14 | / | 179.36 | / | 127.67 | / | 189.50 |
| ESM | 142.79 | 37.80 | 183.84 | 52.44 | 127.50 | 37.47 | 167.50 | 52.40 | 127.00 | 40.00 | 167.00 | 54.25 |
| Py-GAN | / | 28.63 | / | 43.25 | / | 27.30 | / | 41.10 | / | 27.00 | / | 40.88 |
| Lavis | 222.95 | 36.05 | 313.97 | 51.72 | 211.30 | 34.75 | 300.57 | 49.25 | 187.33 | 37.00 | 267.00 | 51.00 |
| EAP | 170.87 | / | 239.68 | / | 121.63 | / | 174.96 | / | 146.20 | / | 205.60 | / |
| If | 243.47 | 160.00 | 325.42 | 201.00 | 272.19 | / | 362.57 | / | 269.33 | / | 361.83 | / |
| vid2vid | / | 85.65 | / | 112.67 | / | 79.85 | / | 104.85 | / | 63.25 | / | 84.75 |
| OpenCLIP | 859.31 | / | 1236.63 | / | 839.55 | / | 1207.91 | / | 913.33 | / | 1313.33 | / |
| TSL | / | 152.98 | / | 205.82 | / | 151.07 | / | 204.71 | / | 152.75 | / | 207.00 |
| **Out-Of-Distribution (OOD)** | | | | | | | | | | | | |
| Py-IM | / | / | / | / | / | 37.40 | / | 53.00 | / | 26.00 | / | 34.00 |
| Learning3D | / | / | / | / | / | 28.59 | / | 41.00 | / | 27.75 | / | 41.00 |
| muzic | / | / | / | / | / | 26.72 | / | 38.72 | / | 14.40 | / | 21.80 |
| Grounded-SAM | / | / | / | / | 177.88 | 48.08 | 271.25 | 67.75 | 177.67 | 62.00 | 271.67 | 88.50 |
| **Average (ID)** | 327.88 | 80.53 | 459.91 | 109.57 | 314.43 | 66.97 | 442.70 | 93.71 | 328.64 | 67.04 | 462.95 | 93.68 |
| **Average (OOD)** | / | / | / | / | 177.88 | 35.20 | 271.25 | 50.12 | 177.67 | 32.54 | 271.67 | 46.33 |
| **Total Average** | 327.88 | 80.53 | 459.91 | 109.57 | 291.79 | 60.12 | 414.09 | 84.15 | 303.64 | 59.04 | 431.07 | 84.11 |

### G.4 DETAILED ANALYSIS OF TASK VARIETY AND INSTRUCTIONAL DEPTH

To provide a clearer understanding of the scope and diversity within ML-BENCH, Table 14 offers a detailed enumeration of the different types of tasks as well as an analysis of the intricacies involved in the instructions that accompany them. Each task category represents a unique section of our dataset, with Multi-Modality tasks taking the lead with 4,732 instances. Time-series and Text-related tasks follow suit with 1,478 and 1,475 instances, respectively, indicating a substantial focus on these areas as well. The numbers are counted by our eight annotators.

Further linguistic analysis revealed the instruction sets' complexity, with an average token length per instruction measuring 80.4 and a maximum token length reaching up to 216 tokens. Additionally, the instruction edit distance—an indicator of linguistic diversity—averages 258.7 tokens within similar tasks and 302.1 tokens across different tasks, underlining the variety and broad coverage of scenarios that ML-BENCH encompasses.

Table 14: Task distribution, instruction complexity, and quantitative analysis of task complexity in ML-BENCH

| Task | Number |
| --- | --- |
| - GNN | 608 |
| - Text | 1475 |
| - Molecular | 649 |
| - Image-GAN | 1189 |
| - Multi-Modality | 4732 |
| - Video | 75 |
| - Time-series | 1478 |
| - Attention Usage | 127 |
| - Medical | 805 |
| - 3D | 264 |
| - Music | 704 |
| **Instruction Complexity** | |
| Average token length per instruction | 80.4 |
| Max token length in instruction | 216 |
| Instruction edit distance among the same task | 258.7 |
| Instruction edit distance across tasks | 302.1 |
| **Code Complexity** | |
| Average number of relevant files per task | 3.6 |
| Average lines of code in relevant files | 414 |
| Average depth of relevant code in repository (line number) | 27,524 |
| Average number of functions/classes to be understood per task | 3.8 |
| **Task Requirements (Percentage of Tasks)** | |
| Requiring README understanding | 85% |
| Requiring code understanding | 95% |
| Requiring understanding of multiple files | 75% |
| Requiring understanding of dependencies | 70% |
| Requiring understanding of data structures | 65% |
| Involving API usage | 80% |

## G.5 DATASET QUALITY ASSESSMENT

To ensure the reliability and relevance of ML-BENCH, we implemented a rigorous quality assurance process during dataset construction and conducted an additional assessment. Our quality control measures include a comprehensive seven-step construction process, as detailed in Section 2.3. This process incorporates a crucial quality assessment step, ensuring that instructions are precisely aligned with user requirements and that all code is executable.

We conducted an additional quality assessment: a. Random sampling: We randomly selected 100 tasks from our dataset for in-depth review. b. Expert evaluation: three senior ML researchers independently reviewed these tasks, assessing their relevance, difficulty, and excitability. c. Execution testing: we ran each selected task through our testing environment to verify its executability and output correctness.

The results of this additional assessment were highly encouraging. 97% of the reviewed tasks were deemed highly relevant to real-world ML workflows, while 95% were successfully executed in our testing environment. Moreover, the average inter-rater agreement on task quality was 0.92 (Cohen's kappa), indicating a high level of consensus among our expert evaluators. These findings strongly support the quality and practical relevance of the ML-BENCH dataset.

## G.6 ERROR ANALYSIS FOR EACH REPOSITORY

Figure 7 illustrates the distribution of errors made by GPT-4 across 14 repositories, categorized as per the error types described in the main text. The analysis was conducted within the context of the ML-BENCH-L, specifically under the Oracle setting.

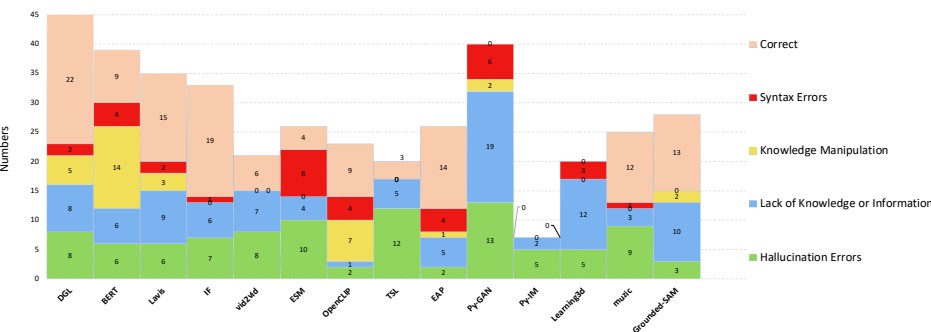

Figure 7: Using the Oracle setup, we ran GPT-4 for five iterations and tallied the number of errors across different repositories to provide an error analysis specific to each repository.

## G.7 ERROR ANALYSIS FOR ML-BENCH-A

Figure 8 illustrates the distribution of errors made by OpenDevin, categorized as per the error types described in the main text. The analysis was conducted within the context of the ML-BENCH-A.

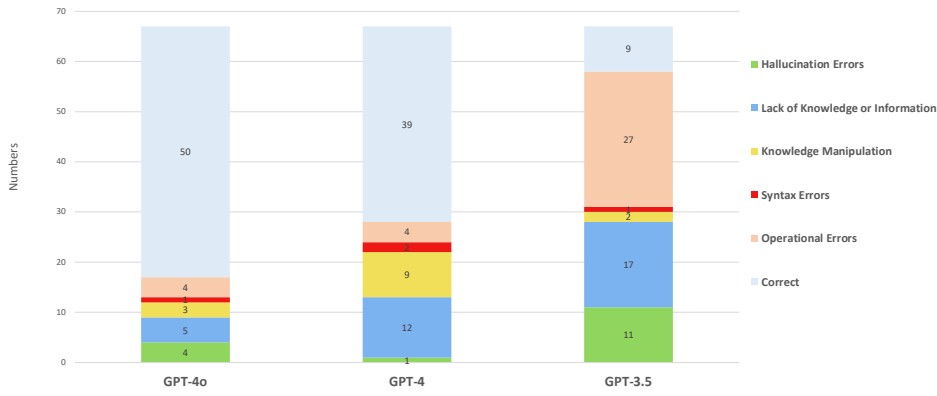

Figure 8: The error analysis for the OpenDevin framework, utilizing various base models on ML-BENCH-A. Notably, Operational Error is a category unique to ML-BENCH-A due to the agents' need to interact dynamically with the coding environment.

## G.8 AN EXAMPLE OF ML-BENCH-L

Figure 9 shows a case from ML-BENCH-L includes a diagram that provides a view of the model's inputs, the reference, multiple outputs generated by the model, and an error analysis with results for each output.

**Model Input**

[readme content]:
DGL is framework agnostic, meaning if a deep graph model is a component of an end-to-end application, the rest of the logics can be implemented in any major frameworks, such as PyTorch, Apache MXNet or TensorFlow....

[instruction]:
I am eager to utilize the *Citeseer* dataset as the training data to empower the ARMA Model with the learning rate set to a commendably small value of *0.0001*. Additionally, I'd like to incorporate *5* stacks into this model. Your assistance in formulating the necessary code to accomplish this task would be of tremendous help.

[System Prompt]:
You are given [readme content], you need to carefully see [readme content] and choose wirte code or script to implement my [instruction].
Please output code or script directly, use markdown to output code without explanation.

**Gold Output**

```
python citation.py --dataset Citeseer --lr 0.0001 --num-stacks 5
```

**Generated Result**

```
"python citation.py --dataset Citeseer --lr 0.01 --stacks 5",

"python main.py --dataset citeseer --lr 0.0001 --stacks 5",

"dgl-go --model=arma --dataset=citeseer --lr=0.0001 --stacks=5",

"import dgl\ndataset = dgl.data.CiteseerGraphDataset()\ng = dataset[0]\nfrom dgllife.model import DAGNNPredictor ...",

"python citation.py --dataset Citeseer --lr 0.0001 --num-stacks 5"
```

**Evaluation Result**

| | | |
|---|---|---|
| **FAILED** | python citation.py --dataset Citeseer --lr 0.01 --stacks 5 | wrong argument |
| **FAILED** | python citation.py --dataset citeseer --lr 0.0001 --stacks 5 | no argument error |
| **FAILED** | dgl-go --model=arma --dataset=citeseer --lr=0.0001 --stacks=5 | no file error |
| **FAILED** | import dgl\ndataset = dgl.data.CiteseerGraphDataset()\n ... | wrong execution |
| **PASSED** | python citation.py --dataset Citeseer --lr 0.0001 --num-stacks 5 | pass |

Figure 9: The illustrative set of **input-output pairs, gold output, and execution results**, accompanied by precision metrics: **Pass@1=0 and Pass@5=1**. Various colors within the instructions signify different parameters.

## G.9 AN EXAMPLE OF ML-BENCH-A

In this section, we present detailed examples of each error type in ML-BENCH-A:

1. Hallucination Errors (E1): Table 15

2. Lack of Knowledge or Information (E2): Table 16

3. Knowledge Manipulation (E3): Table 17

4. Syntax Errors (E4): Table 18

5. Operational Error (E5): Table 19

---

**Instruction:**

I have a digital image titled `example.jpg` that I hope to transform into an ink-wash-styled masterpiece with the IF-I-M-v1.0 model. Can you guide me to write the code to transform the image accordingly?

**Steps:**

**Navigate to the repository**

```
cd /workspace/if/
```

**Attempt to create the `run.sh` script in the wrong directory**

The `run.sh` script is mistakenly attempted to be created in /usr/bin instead of the specified path:

```
echo "#!/bin/bash" > /usr/bin/run.sh
echo "python if_im.py --model_name IF-I-M-v1.0 --input_path example.jpg
    --output_path output.jpg" >> /usr/bin/run.sh
chmod +x /usr/bin/run.sh
```

**Error: Permission denied**

The script creation fails due to insufficient permissions to write in /usr/bin:

```
bash: /usr/bin/run.sh: Permission denied
```

**Error: Script not created and task not executed**

The task is not executed because the script could not be created in the specified path:

```
The run.sh script could not be created in /usr/bin due to permission
    issues. The task in /workspace/if/ was not executed.
```

---

Table 15: An example of E1 error. Task execution attempt for Zero-shot Image-to-Image Translation. The task could not be completed due to an attempt to create the script in /usr/bin, which resulted in permission denial, and the intended task was not executed.

**Instruction:**

Can you assist me in writing the code to extract 5 designs from the structure `Fam20A.pdb`, specifically from chain B, at a temperature of 8?

**Steps:**

**Navigate to the repository**

```
cd /workspace/esm/
```

**Create the `run.sh` script**

The `run.sh` script is created with the following content:

```
#!/bin/bash

python main.py --input Fam20A.pdb --chain B --temperature 8 --output
    designs.txt --num_designs 5
```

**Run the script**

```
./run.sh
```

**Error: Missing `main.py` script**

The execution of `run.sh` fails due to the usage of `main.py` instead of `extract_designs.py`:

```
python: cannot open file main.py: [Errno 2] No such file or directory
```

**Check for the missing script**

The user is asked to ensure that the `main.py` script is present in the repository:

```
Please make sure that the main.py script is available in the /workspace/
    esm/ repository.
```

**Task cannot be completed**

The task cannot be completed without the `main.py` script:

```
I cannot complete the task as requested without the missing script.
```

Table 16: An example of E2 error. Task execution attempt for extracting designs from `Fam20A.pdb`. The task could not be completed due to the misinformation of `main.py` script. The LLM agent should use `extract_designs.py` instead.

**Instruction:**

Looking to use the Cluster GAN model for a project. Hoping to set up a learning rate of 0.0001 and plan on running it for 100 epochs. I would like the batch size to be 16 and to keep the image size at 128. Can you help me craft some code?

**Steps:**

**Create the `run.sh` script**

The `run.sh` script is created with the following content:

```bash
#!/bin/bash

python main.py --model cluster_gan --learning_rate 0.0001 --num_epochs
    100 --batch_size 16 --image_size 128
```

**Run the script**

```
cd /workspace/PyTorch-GAN/implementations/cluster_gan/
chmod +x run.sh
./run.sh
```

**Error: Missing `main.py` script**

The execution of `run.sh` fails due to the missing `main.py` script:

```
python: cannot open file /workspace/PyTorch-GAN/implementations/
    cluster_gan/main.py: [Errno 2] No such file or directory
```

**Update `run.sh` script to use `clustergan.py`**

The `run.sh` script is updated with the following content:

```bash
#!/bin/bash

python clustergan.py --model cluster_gan --learning_rate 0.0001 --
    num_epochs 100 --batch_size 16 --image_size 128
```

**Run the updated script**

```
cd /workspace/PyTorch-GAN/implementations/cluster_gan/
chmod +x run.sh
./run.sh
```

**Error: Unrecognized arguments in `clustergan.py`**

The execution of `run.sh` fails due to unrecognized arguments in `clustergan.py`:

```
usage: clustergan.py [-h] [-n N_EPOCHS] [-b BATCH_SIZE] [-i IMG_SIZE]
                     [-d LATENT_DIM] [-l LEARNING_RATE] [-c N_CRITIC] [-
                     w]
clustergan.py: error: unrecognized arguments: --model cluster_gan --
    learning_rate 0.0001 --num_epochs 100 --image_size 128
```

Table 17: An example of E3 error. Task execution attempt for Cluster GAN model. The task could not be completed due to unrecognized arguments in `clustergan.py`. The LLM agent should have ensured the script accepts the required arguments.

1944
1945
1946
1947
1948
1949
1950
1951
1952
1953
1954
1955
1956
1957
1958
1959
1960
1961
1962
1963
1964
1965
1966
1967
1968
1969
1970
1971
1972
1973
1974
1975
1976
1977
1978
1979
1980
1981
1982
1983
1984
1985
1986
1987
1988
1989
1990
1991
1992
1993
1994
1995
1996
1997

**Instruction:**

Please complete the Machine Learning task in the following repository: esm

The task is: Sample sequences design for certain structure

Can you assist me in writing the code to extract 3 structure examples from the inverse_folding data file `5YH2.pdb`? I specifically need designs with chain A at a temperature of 5. The output should be saved in the `model_structure.fasta` file.

You should create a script named `run.sh` under the specified path in the repo to run the task.

You can find the task repo at: /workspace/esm/

You should terminate the subprocess after running the task (e.g., call `subprocess.Popen(args).wait()`). When you think you have completed the task, please run the following command: <execute_bash> exit </execute_bash>.

**Steps:**

**Create the `run.sh` file using Python code**

Attempt to create the `run.sh` file using Python code fails due to a syntax error:

```
Running Python code interactively: create_file('/workspace/esm/run.sh')

  File "/opendevin/plugins/jupyter/execute_cli.py", line 34
    if jupyter_pwd := os.environ.get('JUPYTER_PWD'):
                   ^
SyntaxError: invalid syntax
```

**Task cannot be completed**

The task cannot be completed due to the syntax error in the Python code:

```
I cannot complete the task as requested due to a syntax error in the
    Python code used to create the run.sh file.
```

Table 18: An example of E4 error. Task execution attempt for Sample sequences design for certain structure. The task could not be completed due to a syntax error in the Python code used to create the `run.sh` file.

| Instruction: |
| --- |
| Can you assist me in writing the code to extract 3 designs from the structure `5YH2.pdb`, specifically from chain B, at a temperature of 1? The output should be saved to the file `model_structure.fasta`. |

| Steps: |
| --- |

**Navigate to the repository**

```
cd /workspace/esm/
```

**Ask the user to create a `run.sh`**

Now let's create the `run.sh` script to extract the designs from the structure. Please create a new file named `run.sh` in the `esm` repository and add the following code to it:

```bash
#!/bin/bash

python extract_designs.py --structure 5YH2.pdb --chain B --temperature 1
    --output model_structure.fasta
```

**Run command**

```
./run.sh
```

```
bash: /workspace/esm/run.sh: No such file or directory
ERROR conda.cli.main_run:execute(47): (*@'conda run bash /workspace/esm/
    run.sh'@*) failed. (See above for error)
```

**Ask the user to create a `run.sh`**

...

**Failed after so many iterations**

```
bash: /workspace/esm/run.sh: No such file or directory
ERROR conda.cli.main_run:execute(47): (*@'conda run bash /workspace/esm/
    run.sh'@*) failed. (See above for error)
```

Table 19: An example of E5 error. Task execution attempt for extracting designs from `5YH3.pdb`. The LLM agent mistakenly asks the human for help and refuses to do the task by itself.

# H EXPERIMENT DETAILS

## H.1 EFFICACY OF BM25 RETRIEVAL ON REPOSITORY-LEVEL CODE

When evaluating the relevance of retrieved code segments within repository-level tasks, we utilized the BLEU-1 score as a metric to measure the similarity between hand-annotated "Oracle" segments and those extracted by the model using BM25 retrieval. "Oracle" segments are those identified by annotators as most pertinent, serving as the foundational evidence required for the task at hand. As depicted in Table 20, the resulting BLEU scores indicate a low degree of correlation, suggesting that the retrieval segments identified by BM25 are significantly dissimilar to the Oracles crafted by human annotators. This finding is demonstrative of BM25's limitations in effectively identifying the most relevant content for repository-scale code, as evidenced by the low BLEU scores.

Table 20: The BLEU-1 scores between Oracles and the Retrieval content.

|  | ID-train | OOD-train | ML-BENCH |
|---|---|---|---|
| BLEU score | 0.0112 | 0.0087 | 0.0082 |

## H.2 INFORMATION LOSSING DUE TO TRUNCATION

It is reasonable that truncation may lead to information missing, but it is worth noting that only in the Code setting for the open-source models does the input of README files need to be truncated to 8k, which is inevitable because of the input length limitation. However, only a small number of README files need to be truncated. To qualitatively present the information loss percentage due to truncation, we present the percentage of losing critical information during truncation in Table 21. Note that all the results are manually examined. We can identify that only 5 repositories lose critical information after truncating the README files.

Table 21: The percentage of losing critical information due to truncation.

| Repos | Proportion of losing information (%) |
|---|---|
| vid2vid | 0 |
| If | 0 |
| DGL | 0 |
| Py-GAN | 33.3 |
| ESM | 11.76 |
| BERT | 100 |
| OpenCLIP | 0 |
| Lavis | 0 |
| TSL | 0 |
| EAP | 75 |
| Grounded-SAM | 0 |
| Py-IM | 20 |
| muzic | 0 |
| Learning3d | 0 |
| SD | 0 |
| MedZooPy | 0 |
| TCL | 0 |
| Tensor2Tensor | 0 |
| Total | 0 |

## H.3 EXAMPLES OF INPUT-OUTPUT OF EACH GITHUB REPOSITORY

In this section, we present detailed examples of the input and output of each GitHub Repo in Tab.22 to Tab.39. The corresponding repository for each table is shown below:

1. External-Attention: Table 22
2. BERT: Table 23
3. Deep learning on graphs: Table 24
4. Evolutionary scale modeling: Table 25
5. Grounded-Segment-Anything: Table 26
6. DeepFloyd IF: Table 27
7. Language-Vision Intelligence: Table 28
8. Deep learning on 3D point clouds data: Table 29
9. 3D multi-modal medical image segmentation library: Table 30
10. Music understanding and generation: Table 31
11. Implementation of OpenAI's CLIP: Table 32
12. Generative Adversarial Network varieties: Table 33
13. PyTorch Image Models: Table 34
14. Stable diffusion: Table 35
15. Text classification: Table 36
16. Tensor2Tensor: Table 37
17. deep time series analysis: Table 38
18. Video-to-video translation: Table 39

**README:**

As a supplement to the project, an object detection codebase, YOLO. Air has recently been opened. It integrates various attention mechanisms in the object detection algorithm. The code is simple and easy to read. Welcome to play and star!

**For beginners (like me):** Recently, I found a problem when reading the paper. Sometimes the core idea of the paper is very simple, and the core code may be just a dozen lines. However, when I open the source code of the author's release, I find that the proposed module is embedded in the task framework such as classification, detection, and segmentation, resulting in redundant code. For me who is not familiar with the specific task framework, it is difficult to find the core code, resulting in some difficulties in understanding the paper and network ideas.

**For advanced (like you):** If the basic units conv, FC, and RNN are regarded as small Lego blocks, and the structures transformer and RESNET are regarded as LEGO castles that have been built, the modules provided by this project are LEGO components with complete semantic information. To avoid repeatedly building wheels, scientific researchers should just think about how to use these "LEGO components" to build more colorful works.

**For proficient (maybe like you):** Limited capacity, do not like light spraying!!!

**For All:** This project aims to realize a code base that can make beginners of deep learning understand and serve scientific research and industrial communities. As fightingcv WeChat official account. The purpose of this project is to achieve Let there be no hard-to-read papers in the world. (at the same time, we also welcome all scientific researchers to sort out the core code of their work into this project, promote the development of the scientific research community, and indicate the author of the code in README)

...

**Oracle Segment:**

..
from model.attention.ViP import WeightedPermuteMLP
import torch
from torch import nn
from torch.nn import functional as F
...

**Instruction:**

I'm planning to utilize the fighting-cv model to complete the attention layers for ViP Attention Usage. Could you provide me with some guidance on accomplishing this task?

**Instruction:**

package_1: ViP
sub_package: WeightedPermuteMLP
package_2: torch

**Ground Truth Output:**

from model.attention.ViP import WeightedPermuteMLP
import torch
from torch import nn
from torch.nn import functional as F
input=torch.randn(64,8,8,512)
seg_dim=8
vip=WeightedPermuteMLP(512,seg_dim)
out=vip(input)
print(out.shape)

Table 22: Example of input-output for **External-Attention-pytorch** GitHub on **attention layer** task on **Attention Usage** domain. The README URL is https://github.com/xmu-xiaoma666/External-Attention-pytorch/blob/master/README_EN.md. The GitHub URL is https://github.com/xmu-xiaoma666/External-Attention-pytorch.

| | |
|---|---|
| **README:** | |

**BERT**

New March 11th, 2020: Smaller BERT Models

This is a release of 24 smaller BERT models (English only, uncased, trained with WordPiece masking) referenced in Well-Read Students Learn Better: On the Importance of Pre-training Compact Models.

...

**Oracle Segment:**

..

This demo code only pre-trains for a smallnumber of steps (20), but in practice you will probably want to set 'num_train_steps' to 10000 steps or more. The 'max_seq_length' and 'max_predictions_per_seq' parameters passed to 'run_pretraining.py' must be the same as 'create_pretraining_data.py'.

```
python run_pretraining.py
    --input_file=/tmp/tf_examples.tfrecord
    --output_dir=/tmp/pretraining_output
    --do_train=True
    --do_eval=True
    --bert_config_file=$BERT_BASE_DIR/bert_config.json
    --init_checkpoint=$BERT_BASE_DIR/bert_model.ckpt
```
...

**Instruction:**

Behold, a formidable quest awaits - the pre-training of the unparalleled uncased_L-24_H-1024_A-16 model. Our path to victory lies in configuring the maximum sequence length to a mighty 512, with a pledge to uphold the limit of 30 predictions per sequence. Battling through the treacherous terrain of 10000 steps, we shall march forward, with a stalwart battalion of 32 batch size by our side. But fear not, for we shall brace ourselves with 10000 warmup steps, as we navigate the perilous sea of learning, with a steadfast learning rate of 0.0001. I humbly beseech your assistance, noble comrade, to conjure the code necessary to conquer this heroic endeavor.

**Arguments Requirements:**

model: ./model/uncased_L-12_H-768_A-16
train_batch_size: 32
max_seq_length: 512
num_train_steps: 10000
num_warmup_steps: 1000
learning_rate: 0.0001

**Ground Truth Output:**

```
python run_pretraining.py
    --input_file=/tmp/tf_examples.tfrecord
    --output_dir=/tmp/pretraining_output
    --do_train=True
    --do_eval=True
    --bert_config_file=./model/uncased_L-24_H-1024_A-16/bert_config.json
    --init_checkpoint=./model/uncased_L-24_H-1024_A-16/bert_model.ckpt
    --train_batch_size=32
    --max_seq_length=512
```
...

Table 23: Example of input-output for **bert** GitHub on **pre-training** task on **BERT** domain. The README URL is https://github.com/google-research/bert/blob/master/README.md. The GitHub URL is https://github.com/google-research/bert.

**README:**

**1. DGL Implementation of CorrectAndSmooth**

This DGL example implements the GNN model proposed in the paper Combining Label Propagation and Simple Models Out-performs Graph Neural Networks. For the original implementation, see here.

Contributor: xnuohz

**2. Requirements**

The codebase is implemented in Python 3.7. For version requirement of packages, see below.

dgl 0.6.0.post1

torch 1.7.0

ogb 1.3.0

...

**Oracle Segment:**

..

**3.1 ogbn-arxiv**

Plain MLP + C&S

```
python main.py
    –dropout 0.5
```

```
python main.py
    --pretrain
    --correction-adj DA
    --smoothing-adj AD
    --autoscale
```

...

**Instruction:**

...

Together, we shall embark on a noble mission to train the illustrious CorrectAndSmooth Model, fortified with a sublime dropout rate of 0.7. Our arduous journey spans 700 epochs, each pulsating with the promise of enlightenment. Alas, I beseech your sage guidance in the ethereal realm of code crafting, to manifest this grand undertaking.

**Arguments Requirements:**

dataset: ogbn-arxiv

model: mlp

dropout: 0.7

epochs: 700

**Ground Truth Output:**

```
python main.py
    --dataset ogbn-arxiv
    --model mlp
    --dropout 0.7
    --epochs 700
```

Table 24: Example of input-output for **DGL** GitHub on **DGL Implementation of CorrectAndSmooth** task on **GNN** domain. The README URL is https://github.com/dmlc/dgl/blob/master/examples/pytorch/correct_and_smooth/README.md. The GitHub URL is https://github.com/dmlc/dgl.

| | |
|---|---|
| 2322 | |
| 2323 | |
| 2324 | |
| 2325 | |
| 2326 | |
| 2327 | |

**README:**

**Evolutionary Scale Modeling**

atlas

Update April 2023: Code for the two simultaneous preprints on protein design is now released! Code for "Language models generalize beyond natural proteins" is under examples/lm-design/. Code for "A high-level programming language for generative protein design" is under examples/protein-programming-language

This repository contains code and pre-trained weights for Transformer protein language models from the Meta Fundamental AI Research Protein Team (FAIR), including our state-of-the-art ESM and ESMFold, as well as MSA Transformer, ESM-1v for predicting variant effects and ESM-IF1 for inverse folding.
...

**Oracle Segment:**

..
The following commands allow the extraction of the final-layer embedding for a FASTA file from the ESM-2 model:

```
esm-extract esm2_t33_650M_UR50D examples/data/some_proteins.fasta
examples/data/some_proteins_emb_esm2
    --repr_layers 0 32 33
    --include
```

```
python scripts/extract.py esm2_t33_650M_UR50D examples/data/some_proteins.fasta
examples/data/some_proteins_emb_esm2
    --repr_layers 0 32 33
    --include mean per_tok
    --A cuda device is optional and will be auto-detected.
```
...

**Instruction:**

...
Can you assist me in writing the code to extract the 24-layer embedding for a FASTA file named rna.fasta using the esm1v_t33_650M_UR90S_5 model and save the output?

**Arguments Requirements:**

model: esm1v_t33_650M_UR90S_5
data: rna.fasta
layer_number: 24
layer_name: repr_layers

**Ground Truth Output:**

```
python scripts/extract.py esm1v_t33_650M_UR90S_5 rna.fasta output.embeddings
    --repr_layers 24
    --include mean per_tok
```

Table 25: Example of input-output for **ESM** GitHub on **Extract ESMFold Structure Prediction Model's Embedding** task on **molecular** domain. The README URL is https://github.com/facebookresearch/esm/blob/master/README.md. The GitHub URL is https://github.com/facebookresearch/esm.

| README: |
| --- |
| Official PyTorch implementation of Grounding DINO), a stronger open-set object detector. Code is available now! |

**Highlight**
- Open-Set Detection. Detect everything with language!
- High Performancce. COCO zero-shot 52.5 AP (training without COCO data!). COCO fine-tune 63.0 AP.
- Flexible. Collaboration with Stable Diffusion for Image Editing.

...

| Oracle Segment: |
| --- |
| .. |

Demo
python demo/inference_on_a_image.py
  -c /path/to/config
  -p /path/to/checkpoint
  -i .asset/cats.png
  -o outputs/0
  -t cat ear. [--cpu-only] # open it for cpu mode
See the demo/inference_on_a_image.py for more details.

...

| Instruction: |
| --- |
| ... |

I am interested in utilizing the grounding dino demo for a specific task. The input image path is ground_segment/GD_new.json, and I would like the output to be saved in the directory output/-cat2002. Additionally, I would like the text condition to be set to right ear of cat. Could you kindly assist me in writing the script to achieve this?

| Arguments Requirements: |
| --- |
| i: .asset/cat.jpg |

o: output/cat2002
t: right ear of cat

| Ground Truth Output: |
| --- |
| python demo/inference_on_a_image.py |

  -c model/GroundingDINO_SwinT_OGC.py
  -p model/groundingdino_swint_ogc.pth
  -i .asset/cat.jpg
  -o output/cat2002
  -t right ear of cat

Table 26: Example of input-output for **Grounded-Segment-Anything** GitHub on **Grounding DINO demo** task on **Segment** domain. The README URL is https://github.com/IDEA-Research/Grounded-Segment-Anything/blob/main/GroundingDINO/README.md. The GitHub URL is https://github.com/IDEA-Research/Grounded-Segment-Anything.

| | |
|---|---|
| 2430 | |
| 2431 | |
| 2432 | |
| 2433 | |
| 2434 | |

**README:**

We introduce DeepFloyd IF, a novel state-of-the-art open-source text-to-image model with a high degree of photorealism and language understanding. DeepFloyd IF is a modular composed of a frozen text encoder and three cascaded pixel diffusion modules: a base model that generates 64x64 px image based on text prompt and two super-resolution models, each designed to generate images of increasing resolution: 256x256 px and 1024x1024 px. All stages of the model utilize a frozen text encoder based on the T5 transformer to extract text embeddings, which are then fed into a UNet architecture enhanced with cross-attention and attention pooling. The result is a highly efficient model that outperforms current state-of-the-art models, achieving a zero-shot FID score of 6.66 on the COCO dataset. Our work underscores the potential of larger UNet architectures in the first stage of cascaded diffusion models and depicts a promising future for text-to-image synthesis.

...

**Oracle Segment:**

..

**II. Zero-shot Image-to-Image Translation**

In Style Transfer mode, the output of your prompt comes out at the style of the support_pil_img

from deepfloyd_if.pipelines import style_transfer

result = style_transfer(
t5 = t5, if_I = if_I, if_II = if_II,
...

**Instruction:**

...
Time to create a visual masterpiece! I am excited to recreate 'image'.jpg into a fascinating rendition of the ink wash style. I am planning to utilize the capabilities of the IF-I-XL-v1.0 model for this endeavor. Can you assist me in crafting the necessary code?

**Arguments Requirements:**

model: IF-I-XL-v1.0
argument1: image.jpg
argument2: wash

**Ground Truth Output:**

from deepfloyd_if.modules import IFStageI, IFStageII, StableStageIII
from deepfloyd_if.modules.t5 import T5Embedder
device = 'cuda:1'
if_I = IFStageI('IF-I-XL-v1.0', device=device)
if_II = IFStageII('IF-II-L-v1.0', device=device)
if_III = StableStageIII('stable-diffusion-x4-upscaler', device=device)
t5 = T5Embedder(device='cpu')

style_prompt = a captivating ink wash style
image_path = image.jpg
...

Table 27: Example of input-output for **IF** GitHub on **Zero-shot Image-to-Image Translation** task on **Image and Text** domain. The README URL is https://github.com/deep-floyd/IF/blob/develop/README.md. The GitHub URL is https://github.com/deep-floyd/if.

**README:**

**Lavis**

Lavis - A Library for Language-Vision Intelligence

What's New:

A simple, yet effective, cross-modality framework built atop frozen LLMs that allows the integration of various modalities (image, video, audio, 3D) without extensive modality-specific customization.

Technical Report and Citing LAVIS:

...

**Oracle Segment:**

...

how to use models in LAVIS to perform inference on example data. We first load a sample image from local.

import torch

from PIL import Image

# setup device to use

device = torch.device("cuda" if torch.cuda.is_available() else "cpu")

# load sample image

raw_image = Image.open(merlion.png).convert(RGB)

This example image shows Merlion park (source), a landmark in Singapore.

# Image Captioning

In this example, we use the BLIP model to generate a caption for the image.

...

**Instruction:**

There is a task in front of me currently, which is the Image Captioning task. I ask for your kind help in moving forward with this task.

**Arguments Requirements:**

image: image.jpg

**Ground Truth Output:**

```
import torch
from PIL import Image
from lavis.models import load_model_and_preprocess
device = torch.device('cuda:1' if torch.cuda.is_available() else 'cpu')
model, vis_processors, _ = load_model_and_preprocess(name='blip_caption',
model_type='base_coco', is_eval=True, device=device)
raw_image = Image.open('image.jpg').convert('RGB')
preprocessed_image = vis_processors['eval'](raw_image).unsqueeze(0).to(device)
output = model.generate({'image': preprocessed_image})
caption = output['captions'][0]['text']
print(caption)
```

Table 28: Example of input-output for **Lavis** GitHub on **Image Captioning** task on **Multimodal Image and Text** domain. The README URL is https://github.com/salesforce/LAVIS/blob/main/README.md. The GitHub URL is https://github.com/salesforce/lavis.

**README:**

**Learning3D: A Modern Library for Deep Learning on 3D Point Clouds Data.**
Learning3D is an open-source library that supports the development of deep learning algorithms that deal with 3D data. The Learning3D exposes a set of state of art deep neural networks in python. A modular code has been provided for further development. We welcome contributions from the open-source community.

**Available Computer Vision Algorithms in Learning3D**

...

**Oracle Segment:**

...
examples/test_dcp.py Learning3D is an open-source library that supports the development of deep learning algorithms that deal with 3D data. The Learning3D exposes a set of state of art deep neural networks in python

```
python test_dcp.py
    --num_points 128
    --j 12
    --symfn max
```
...

**Instruction:**

I am interested in conducting a test using the dcp model. Specifically, I would like to set the parameters as follows: the test mode should be selected, the model should be set to dcp, the number of points should be 512, the number of data loading workers should be -j 8, and the symmetric function should be set to –symfn max. Could you please assist me in writing the code or script necessary to carry out this test?

**Arguments Requirements:**

number of points: 512
number of data loading workers: 8
symmetric function: max

**Ground Truth Output:**

```
python test_dcp.py
    --num_points 512
    --j 8
    --symfn max
```

Table 29: Example of input-output for **Learning3D** GitHub on **Test dcp model** task on **3D** domain. The README URL is https://github.com/vinits5/learning3d/blob/master/README.md. The GitHub URL is https://github.com/vinits5/learning3d.

| | |
|---|---|
| **README:** | |
| **MusicBERT** | |

Basics

All models accept two parameters: a) the input the channels (in_channels), and b) the segmentation classes (classes) and produce un-normalized outputs

All losses accept as input the prediction in 5D shape of [batch,classes,dim_1,dim_2,dim_3] and the target in 4D target shape of [batch, dim_1, dim_2, dim_3]. It is converted to one-hot inside the loss function for consistency reasons.

Furthermore the normalization of the predictions is handled here. Dice-based losses return the scalar loss for backward(), and the prediction per channels in numpy to track training progress.

...

**Oracle Segment:**

Usage

How to train your model

For Iseg-2017 :

python ./examples/train_iseg2017_new.py
    --args

For MR brains 2018 (4 classes)

python ./examples/train_mrbrains_4_classes.py
    --args

For MR brains 2018 (8 classes)

python ./examples/train_mrbrains_9_classes.py
    --args

For MICCAI 2019 Gleason Challenge

python ./examples/test_miccai_2019.py
    --args

The arguments that you can modify are extensively listed in the manual.

**Instruction:**

I'm seeking assistance in writing a piece of code that can successfully train a model for the 'Iseg 2017 Task'. The model in question is 'RESNET3DVAE' and I require the learning rate to be set to '1e-3'. It is also crucial that the training samples are set to '10'. Lastly, use 'sgd' as the optimizer. Could you kindly help out in creating this algorithm?

**Arguments Requirements:**

lr: 1e-3
samples_train: 10
model: RESNET3DVAE
soptimizer: sg

**Ground Truth Output:**

python ./examples/train_iseg2017_new.py
    --lr 1e-3
    --samples_train 10
    --model RESNET3DVAE
    --opt sgd

Table 30: Example of input-output for **MedicalZoo** GitHub on **Iseg-2017** task on **Medical** domain. The README URL is https://github.com/black0017/MedicalZooPytorch/blob/master/manual/README.md. The GitHub URL is https://github.com/black0017/MedicalZooPytorch.

| |
|---|
| **README:** |
| **MusicBERT** |
| MusicBERT: Symbolic Music Understanding with Large-Scale Pre-Training, by Mingliang Zeng, Xu Tan, Rui Wang, Zeqian Ju, Tao Qin, Tie-Yan Liu, ACL 2021, is a large-scale pre-trained model for symbolic music understanding. It has several mechanisms including OctupleMIDI encoding and bar-level masking strategy that are specifically designed for symbolic music data, and achieves state-of-the-art accuracy on several music understanding tasks, including melody completion, accompaniment suggestion, genre classification, and style classification. |
| Projects using MusicBERT: |
| midiformers: a customized MIDI music remixing tool with easy interface for users. |
| **1. Preparing datasets** |
| **1.1 Pre-training datasets** |
| Prepare |
| tar -xzvf lmd_full.tar.gz |
| zip -r lmd_full.zip lmd_full |
| Run the dataset processing script. ('preprocess.py') |
| python -u preprocess.py |
| The script should prompt you to input the path of the midi zip and the path for OctupleMIDI output. |
| ... |
| **Oracle Segment:** |
| Pre-training bash train_mask.sh lmd_full small Download our pre-trained checkpoints here: small and base, and save in the checkpoints folder. (a newer version of fairseq is needed for using provided checkpoints: see issue-37 or issue-45) |
| **Instruction:** |
| I am interested in conducting a test using the dcp model. Specifically, I would like to set the parameters as follows: the test mode should be selected, the model should be set to dcp, the number of points should be 512, the number of data loading workers should be -j 8, and the symmetric function should be set to –symfn max. Could you please assist me in writing the code or script necessary to carry out this test? |
| **Arguments Requirements:** |
| bash: train_mask.sh |
| dataset: lmd_full |
| checkpoint: small |
| **Ground Truth Output:** |
| bash train_mask.sh lmd_full small |

Table 31: Example of input-output for **Muzic** GitHub on **Pre-training model** task on **Music** domain. The README URL is https://github.com/microsoft/muzic/blob/main/musicbert/READM E.md. The GitHub URL is https://github.com/microsoft/muzic.

2700
2701
2702
2703
2704
2705
2706
2707

| | |
|---|---|
| **README:** | |

**OpenCLIP**

Welcome to an open-source implementation of OpenAI's CLIP(Contrastive Language-Image Pre-training).

Using this codebase, we have trained several models on a variety of data sources and compute budgets, ranging from small-scale experiments to larger runs including models trained on datasets such as LAION-400M, LAION-2B and DataComp-1B.

Many of our models and their scaling properties are studied in detail in the paper reproducible scaling laws for contrastive language-image learning.

Some of our best models and their zero-shot ImageNet-1k accuracy are shown below, along with the ViT-L model trained by OpenAI. We provide more details about our full collection of pretrained models here, and zero-shot results for 38 datasets here.

midiformers: a customized MIDI music remixing tool with easy interface for users.

...

**Oracle Segment:**

Pre-training bash train_mask.sh lmd_full small Download our pre-trained checkpoints here: small and base, and save in the checkpoints folder. (a newer version of fairseq is needed for using provided checkpoints: see issue-37 or issue-45)

**Instruction:**

I'm hoping you could assist me with a task. I need to sort a collection of images located at the directory '. /image'. For this purpose, I'll be using the 'ViT-B-32' model along with the pretraining model 'commonpool_s_laion_s13m_b4k'. Would you be able to provide instructions or steps on how to accomplish this task?

**Arguments Requirements:**

model: ViT-B-32
pretraining_model: commonpool_s_laion_s13m_b4k
path: ./image

**Ground Truth Output:**

```
import os
import numpy as np
import torch
import open_clip
open_clip.list_pretrained()
model, _, preprocess =
open_clip.create_model_and_transforms('ViT-B-32', pretrained='commonpool_s_laion_s13m_b4k')
model.eval()
context_length = model.context_length
vocab_size = model.vocab_size
...
```

Table 32: Example of input-output for **OpenCLIP** GitHub on **Image classification** task on **Image and Text** domain. The README URL is https://github.com/mlfoundations/open_clip/blob/main/README.md. The GitHub URL is https://github.com/mlfoundations/open_clip.

2744
2745
2746
2747
2748
2749
2750
2751
2752
2753

**README:**

**MusicBERT**

Collection of PyTorch implementations of Generative Adversarial Network varieties presented in research papers. Model architectures will not always mirror the ones proposed in the papers, but I have chosen to focus on getting the core ideas covered instead of getting every layer configuration right. Contributions and suggestions of GANs to implement are very welcomed.

Table of Contents:

    Installation

    Implementations

    Auxiliary Classifier GAN

    Adversarial Autoencoder

...

**Oracle Segment:**

...

Energy-Based GAN

Among them, we show one instantiation of EBGAN framework as using an auto-encoder architecture, with the energy being the reconstruction error, in place of the discriminator. We show that this form of EBGAN exhibits more stable behavior than regular GANs during training. We also show that a single-scale architecture can be trained to generate high-resolution images.

Run Example

$ cd implementations/ebgan/

$ python3 ebgan.py

**Instruction:**

I have a task to work with the Energy-Based GAN model. The learning rate for this task needs to be set at 0.0001, the number of training epochs should be defined as 100, and the batch size should be fixed at 16. Furthermore, I want the image size to be set at 128. Can you please assist me in framing the script to facilitate this?

**Arguments Requirements:**

lr: 0.0001

n_epochs: 100

batch_size: 16

img_size: 128

model: ebgan

**Ground Truth Output:**

```
python3 ebgan.py
    --lr 0.0001
    --n_epochs 100
    --batch_size 16
    --mg_size 128
```

Table 33: Example of input-output for **pyGAN** GitHub on **Energy-Based GAN** task on **images-many-GANs** domain. The README URL is https://github.com/eriklindernoren/PyTorch -GAN/blob/master/README.md. The GitHub URL is https://github.com/eriklindernoren/P yTorch-GAN.

**README:**

**PyTorch Image Models**

...

**What's new**

...

**Introduction**

PyTorch Image Models (timm) is a collection of image models, layers, utilities, optimizers, schedulers, data-loaders / augmentations, and reference training / validation scripts that aim to pull together a wide variety of SOTA models with ability to reproduce ImageNet training results.

...

**Oracle Segment:**

..

Existing method of changing patch_size (resize pretrained patch_embed weights once) on creation still works.

**Example validation cmd**

python validate.py /imagenet
   --model vit_base_patch16_224
   --amp
   --amp-dtype bfloat16
   --img-size 255
   --crop-pct 1.0
   --model-kwargs dynamic_img_size=True dyamic_img_pad=True

...

**Instruction:**

I am interested in performing the task of resizing the image or window. For this purpose, I would like to utilize the model vit_base_patch16_224. Additionally, it would be helpful to set the amp-dtype to bfloat16. Moreover, I would like to specify the image size as 255 and the crop percentage as 1.0. To ensure flexibility, I would like to enable dynamic image size and dynamic image padding. Could you kindly assist me in creating the code or script to accomplish this objective?

**Arguments Requirements:**

model: vit_base_patch16_224
amp-dtype: bfloat16
img-size: 255
crop-pct: 1.0
dynamic_img_size: True
dyamic_img_pad: True

**Ground Truth Output:**

python validate.py /imagenet
   --model vit_base_patch16_224
   --amp
   --amp-dtype bfloat16
   --img-size 255
   --crop-pct 1.0
   --model-kwargs dynamic_img_size=True

Table 34: Example of input-output for **PyIM** GitHub on **PyIM Implementation of Resize The Image/Window** task on **Image** domain. The README URL is https://github.com/huggingface/pytorch-image-models/blob/main/README.md. The GitHub URL is https://github.com/huggingface/pytorch-image-models.

**README:**

**Stable Diffusion Version 2**

This repository contains Stable Diffusion models trained from scratch and will be continuously updated with new checkpoints. The following list provides an overview of all currently available models. More coming soon.

...

**Requirements**

You can update an existing latent diffusion environment by running.

...

**Oracle Segment:**

...

We provide the configs for the SD2-v (768px) and SD2-base (512px) model. First, download the weights for SD2.1-v and SD2.1-base. To sample from the SD2.1-v model, run the following:

python scripts/txt2img.py
    --prompt "a professional photograph of an astronaut riding a horse"
    --ckpt <path/to/768model.ckpt\>
    --config configs/stable-diffusion/v2-inference-v.yaml
    --H 768
    --W 768

or try out the Web Demo: Hugging Face Spaces.

...

**Instruction:**

...

For the task of generating an image from text, I need your assistance in writing the code. We'll be using the scripts/txt2img.py script along with the SD2.1-v model. Ensure that the model checkpoint file is located at ......  As we want to generate a high-quality image, set the number of sampling steps to 20. The prompt to generate the image is "a professional photograph of an astronaut riding a horse" and we only need one iteration of the generation process. Can you help me write the code to accomplish this task?

**Arguments Requirements:**

repeat: 1
config: "configs/stable-diffusion/v2-inference-v.yaml"
ckpt: "ckpt/SD2_1_v_model.ckpt"
prompt: "a professional photograph of an astronaut riding a horse"
precision: full
steps: 20
seed: 2048

**Ground Truth Output:**

python scripts/txt2img.py
    --prompt "a professional photograph of an astronaut riding a horse"
    --ckpt ckpt/SD2_1_v_model.ckpt
    --config configs/stable-diffusion/v2-inference-v.yaml
    --H 768
    --W 768
    --seed 2048
    --precision full
    --steps 20
    --repeat 1

Table 35: Example of input-output for **SD** GitHub on **SD Implementation of Text-to-Image** task on **Stable Diffusion** domain. The README URL is https://github.com/Stability-AI/stablediffusion/blob/main/README.md. The GitHub URL is https://github.com/Stability-AI/stablediffusion.

**README:**

**Text Classification**
The purpose of this repository is to explore text classification methods in NLP with deep learning.
...
**Usage:**
1.model is in xxx_model.py
2.run python xxx_train.py to train the model
...

**Oracle Segment:**
it learn represenation of each word in the sentence or document with left side context and right side context:
representation current word=[left_side_context_vector,current_word_embedding,right_side_context_vecotor]. for left side context, it use a recurrent structure, a no-linearity transfrom of previous word and left side previous context; similarly to right side context.check: p71_TextRCNN_model.py

**Instruction:**
I am looking to utilize the TextRCNN model for a particular task. In the course of executing this task, I would like to fix the learning rate at 0.00001, the number of training epochs at 300, and set my batch size to 16. Are you in a position to assist me in creating the appropriate coding syntax for this purpose?

**Arguments Requirements:**
model: TextRCNN
learning_rate: 0.00001
num_epochs: 300
batch_size: 16

**Ground Truth Output:**
python3 a04_TextRCNN/p71_TextRCNN_train.py
 --num_epochs 300
 --batch_size 16
 --lr 0.00001

Table 36: Example of input-output for **TC** GitHub on **TC Implementation of TextRCNN** task on **texts** domain. The README URL is https://github.com/brightmart/text_classificatio n/blob/master/README.md. The GitHub URL is https://github.com/brightmart/text_cla ssification.

| README: |
| --- |

**Tensor2Tensor**

Tensor2Tensor, or T2T for short, is a library of deep learning models and datasets designed to make deep learning more accessible and accelerate ML research.

...

**Quick Start**

This iPython notebook explains T2T and runs in your browser using a free VM from Google, no installation needed. Alternatively, here is a one-command version that installs T2T, downloads MNIST, trains a model and evaluates it:

...

| Oracle Segment: |
| --- |

...
Test the quick-start on a Workspace's Terminal with this command
t2t-trainer –generate_data
  --data_dir=./t2t_data
  --output_dir=./t2t_train/mnist
  --problem=image_mnist
  --model=shake_shake
  --hparams_set=shake_shake_quick
  --train_steps=1000
  --eval_steps=100
Note: Ensure compliance with the FloydHub Terms of Service.

...

| Instruction: |
| --- |

I'd like assistance in trying to make sense of mathematical language. Any help you can give would be highly appreciated, in comprehending the intricate nature of mathematical terminologies and symbols.

| Ground Truth Output: |
| --- |

t2t-trainer –generate_data
  --data_dir= /t2t_data
  --output_dir= /t2t_train/mathematical_language
  --problem=algorithmic_math_two_variables
  --model=transformer
  --hparams_set=transformer_tiny
  --train_steps=1000
  --eval_steps=100

Table 37: Example of input-output for **tensor2** GitHub on **tensor2 Implementation of Mathematical Language Understanding** task on **Text** domain. The README URL is https://github.com/tensorflow/tensor2tensor/blob/master/README.md. The GitHub URL is https://github.com/tensorflow/tensor2tensor.

| README: |
| --- |
| **Time Series Library (TSlib)** |
| TSlib is an open-source library for deep learning researchers, especially for deep time series analysis. |
| We provide a neat code base to evaluate advanced deep time series models or develop your model, which covers five mainstream tasks: long- and short-term forecasting, imputation, anomaly detection, and classification. |
| ... |
| **Usage** |
| 1.Install Python 3.8. For convenience, execute the following command. |
| pip install -r requirements.txt |
| ... |

| Oracle Segment: |
| --- |
| ... |
| Train and evaluate model. We provide the experiment scripts for all benchmarks under the folder ./scripts/. You can reproduce the experiment results as the following examples: |
| **long-term forecast** |
| bash ./scripts/long_term_forecast/ETT_script/TimesNet_ETTh1.sh |
| **short-term forecast** |
| bash ./scripts/short_term_forecast/TimesNet_M4.sh |
| ... |

| Instruction: |
| --- |
| I'm aiming to conduct a performance evaluation of my FEDformer model which is designated for a classification task. I would like to utilize subset: FaceDetection of the UEA benchmark dataset for this evaluation. The patience parameter needs to be set to 10, with the train epochs set to 16, and batch size adjusted to 100. Could you kindly provide me the code necessary to accomplish this? |

| Arguments Requirements: |
| --- |
| task_name: classification |
| model: FEDformer |
| data: UEA |
| model_id: FaceDetection |
| patience: 10 |
| batch_size: 16 |
| train_epochs: 100 |

| Ground Truth Output: |
| --- |
| python -u run.py |
|    --task_name classification |
|    --model FEDformer |
|    --data UEA |
|    --e_layers 3 |
|    --batch_size 16 |
|    --d_model 128 |
|    --d_ff 256 |
|    --top_k 3 |
|    --train_epochs 100 |
|    --patience 10 |
| ... |

Table 38: Example of input-output for **tsl** GitHub on **tsl Implementation of classification** task on **Time-series** domain. The README URL is https://github.com/thuml/Time-Series-Library/blob/main/README.md. The GitHub URL is https://github.com/thuml/Time-Series-Library.

**README:**

**vid2vid**

...

Pytorch implementation for high-resolution (e.g., 2048x1024) photorealistic video-to-video translation. It can be used for turning semantic label maps into photo-realistic videos, synthesizing people talking from edge maps, or generating human motions from poses. The core of video-to-video translation is image-to-image translation. Some of our work in that space can be found in pix2pixHD and SPADE.

...

**Getting Started**

...

**Oracle Segment:**

...

To test the model (bash ./scripts/face/test_512.sh): #!./scripts/face/test_512.sh

python test.py
    --name edge2face_512
    --dataroot datasets/face/
    --dataset_mode face
    --input_nc 15
    --loadSize 512
    --use_single_G

The test results will be saved in: ./results/edge2face_512/test_latest/.

**Instruction:**

We have the task of training the edge2face model on a face dataset. Let's set the resolution size this time to 256. Can you provide assistance in drafting the applicable coding structure?

**Arguments Requirements:**

model: edge2face
dataset: face
resolusion size: 256
argument: g1

**Ground Truth Output:**

python train.py
    --name edge2face_256_g1
    --dataroot datasets/face/
    --dataset_mode face
    --input_nc 15
    --loadSize 256
    --ngf 64
    --max_frames_per_gpu 6
    --n_frames_total 12
    --niter 20
    --niter_decay 20

Table 39: Example of input-output for **vid2vid** GitHub on **vid2vid Implementation of Edge to Face Training** task on **Videos** domain. The README URL is https://github.com/NVIDIA/vid2vid/blob/master/README.md. The GitHub URL is https://github.com/NVIDIA/vid2vid.

## H.4 BM25

**Retrieval**: In this setting, we apply BM25 retriever to get the instruction-related segment in README files. We set the BM25 retriever's retrieval span at 10 sentences using the NLTK sentence tokenizer because the average length of an Oracle is 9.5.

## H.5 RESULTS ACROSS DIFFERENT GITHUB REPOSITORIES

Table 40: This table shows the Pass@5 scores of GPT-4, and Claude on the ¼ set (*) and GPT 3.5 and Claude on the full set across different GitHub repositories. **Oracle**, **Code** and **BM25** represents Oracle, Code, and Retrieval settings.

| Repository | GPT-4* | | | Claude * | | | GPT 3.5 | | | Claude | | |
|---|---|---|---|---|---|---|---|---|---|---|---|---|
| | Oracle | Code | BM25 | Oracle | Code | BM25 | Oracle | Code | BM25 | Oracle | Code | BM25 |
| DGL | **80.00** | 60.00 | 60.00 | 40.00 | 20.00 | 80.00 | **47.62** | 23.81 | 23.81 | 28.57 | 19.05 | 14.29 |
| BERT | 50.00 | 50.00 | 16.67 | 0.00 | **80.00** | 16.67 | **22.73** | 13.63 | 13.63 | 0.00 | 4.54 | 0.00 |
| Lavis | 42.86 | 71.43 | 42.86 | 57.14 | **85.71** | 14.29 | 55.56 | **70.37** | 51.85 | 51.85 | 59.26 | 29.63 |
| If | **100.00** | **100.00** | 33.33 | **100.00** | 0.00 | 13.33 | 71.43 | 61.90 | 52.38 | 71.43 | **76.19** | 52.38 |
| vid2vid | 50.00 | **75.00** | 50.00 | 0.00 | 25.00 | 50.00 | **92.31** | 76.92 | 69.23 | 76.92 | 38.46 | 15.38 |
| ESM | 60.00 | 0.00 | 80.00 | 0.00 | **100.00** | 20.00 | 47.06 | 29.41 | **58.82** | 5.88 | 11.76 | 11.76 |
| OpenCLIP | **66.67** | **66.67** | **66.67** | **66.67** | **66.67** | 0.00 | **63.63** | 36.36 | 54.55 | **63.63** | **63.63** | 45.46 |
| TSL | 25.00 | 25.00 | 0.00 | 25.00 | 0.00 | 0.00 | **14.29** | **14.29** | 0.00 | 7.14 | 7.14 | 0.00 |
| EAP | **100.00** | 80.00 | 0.00 | 100.00 | 20.00 | 80.00 | 66.66 | 70.83 | 33.33 | 70.83 | **83.33** | 20.83 |
| Py-GAN | 0.00 | **12.50** | 0.00 | 0.00 | **12.50** | 0.00 | **6.67** | 0.00 | 0.00 | 0.00 | 0.00 | 0.00 |
| Py-IM | 0.00 | 0.00 | 0.00 | 0.00 | 0.00 | 0.00 | **20.00** | 0.00 | 0.00 | 0.00 | 0.00 | 0.00 |
| Learning3d | 0.00 | 0.00 | 0.00 | 25.00 | 0.00 | 25.00 | 23.53 | **47.06** | 35.29 | 17.65 | 0.00 | 0.00 |
| muzic | **80.00** | 60.00 | 40.00 | 60.00 | 20.00 | 20.00 | 66.67 | **72.22** | 61.11 | 38.89 | 33.33 | 33.33 |
| Grounded-SAM | **60.00** | **60.00** | 20.00 | 0.00 | 0.00 | 0.00 | 0.00 | **20.00** | 0.00 | 5.00 | 35.00 | 10.00 |
| Total | **48.53** | 45.59 | 27.94 | 34.25 | 35.61 | 20.55 | 36.92 | 35.39 | 22.69 | 30.38 | 32.31 | 16.92 |

# I RELATED WORK

## I.1 CODE GENERATION

Code generation in natural language processing (NLP) has been a significant research topic, leading to the development of various methodologies and benchmarks, as seen in (Cassano et al., 2022; Chen et al., 2021; Christopoulou et al., 2022; Li et al., 2022; Orlanski et al., 2023; Tang et al., 2023a;b; Wang et al., 2023b). Current benchmarks primarily aim to enhance function-level code generation capabilities. However, ML-BENCH diverges by integrating code generation to streamline the usage of repositories within real-world workflows. For a comparative overview, see Table 41. The goal of function-level code generation is the creation of code snippets tailored to user needs or to augment code completion processes (Feng et al., 2020; Li et al., 2022), which includes the development of code LLMs (Bi et al., 2024; Zheng et al., 2023).

Table 41: Comparison of benchmarks: characterizing existing function-level benchmarks and ML-BENCH.

| Bench Name | Type | Language | # Samples |
|---|---|---|---|
| ML-Bench | Task Execution | Python & Bash | 9,641 |
| HumanEval (Chen et al., 2021) | Function Completion | Python | 164 |
| MBPP (Austin et al., 2021) | Function Completion | Python | 1,000 |
| DS-1000 (Lai et al., 2023) | Function Completion | Python | 1,000 |
| APPS (Hendrycks et al., 2021a) | Function Completion | Python | 10,000 |

## I.2 AGENT

The integration of AI agents in software development is rapidly advancing, with projects like OpenDevin (Wang et al., 2024b), SWE-agent (Yang et al., 2024), AutoGen (Wu et al., 2023), and Aider (Gauthier) showcasing diverse methodologies for augmenting developers' capabilities. OpenDevin[§] leverages open-source technologies to automate coding and debugging, thereby streamlining development workflows. SWE-agent's ACI allows language models to independently tackle software engineering tasks, exhibiting impressive outcomes in benchmark tests. AutoGen's collaborative agent framework melds conversational AI with human and digital tools to automate a breadth of tasks, from programming to problem-solving. Finally, Aider brings LLMs directly into the coding process, enabling true co-editing experiences between AI models like GPT-4o, Claude 3 Opus, and developers within git repositories, enhancing code editing and project management.

