# OpenReview forum: "ML-Bench: Evaluating Large Language Models for Code Generation in Repository-Level Machine Learning Tasks"
_ICLR.cc/2025/Conference — Submitted to ICLR 2025_

### Official Review · Reviewer_gnam · 2024-11-02

**Soundness:** 2
**Presentation:** 3
**Contribution:** 3
**Rating:** 6
**Confidence:** 3

**Summary:**

This paper presents ML-Bench as a new repository-level software engineering benchmark that targets ML applications. One could see ML-Bench as an ML counterpart for SWE-Bench. For example, ML-Bench provides two interfaces for both LLM-based text-to-code evaluation (ML-LLM-Bench) and sandboxes (ML-Agent-Bench) that can provide feedback for agent frameworks to evaluate their real-world problem-solving abilities.

**Strengths:**

1. The writing is easy to follow with self-explanatory illustrations.
2. One difference between setting agents for ML applications and general software applications is that ML applications usually require comprehensive dependencies and dataset preparation; ML-Bench puts some effort into this part to highlight the necessity to build benchmarks towards ML applications.
3. According to L264 to L269, ML-Bench has in-distribution (ID) and out-of-distributions (OOD) evaluation cases. The results show that even ID fine-tuning can not bring statistically obvious improvement for some models.

**Weaknesses:**

1. Missing details about Success Rate evaluation in ML-Agent-Bench. I understand it should be up to the specific agent framework to evaluate whether an instance is solved, but how do you define if an agent execution fulfills user requirements, especially for ML applications? Unlike the general SWE domain, ML repositories are usually not accompanied by heavy unit tests, and even if they do, due to the stochastic nature of ML applications, it's hard to say an instance is not solved merely because the final output does not match the pre-defined results.

2. I see the authors have extensively discussed the difference between SWE-Bench and ML-Bench. Still, I do not agree with some of the claims in Appendix B.1. For example, the c) claims repository understanding, which is also the focus of SWE-Bench. e) It claims SWE-Bench does not "explicitly" address the documentation utilization, where I consider documentation to be easily considered pure-text code files in both agent and agent-less solutions.

**Questions:**

Refer to the Weaknesses.

**Details Of Ethics Concerns:**

I would like to flag this submission for further ethics review. I noticed the authors asked "eight computer science graduate students with proficient programming abilities" to annotate 18 ML repositories; each of them spent over 30 hours constructing the data. However, no IRB approval or fair payment information was disclosed in this paper. It might be fine if these students were listed as authors. Otherwise, IRB and fair compensation information might be mandatory for this amount of human effort.

---

> ### Author Response · Authors · 2024-11-12
> **Ethics concerns about data annotation**
>
> > I would like to flag this submission for further ethics review. I noticed the authors asked "eight computer science graduate students with proficient programming abilities" to annotate 18 ML repositories; each of them spent over 30 hours constructing the data. However, no IRB approval or fair payment information was disclosed in this paper. It might be fine if these students were listed as authors. Otherwise, IRB and fair compensation information might be mandatory for this amount of human effort.
>
>
>
> Dear Reviewer gnam,
>
> Thank you for your thorough and constructive review of our paper. Before addressing your two main concerns regarding the Success Rate evaluation and the comparison with SWE-Bench, we would like to clarify the data annotation process that you raised in your ethics concerns:
>
> **The eight computer science graduate students who participated in the data annotation process are actually co-authors of this paper.** We apologize for not making this explicit in the manuscript. These students were integral members of our research team and contributed significantly to both the dataset construction and other aspects of the research, like paper writing and experiments. This is why they are included as authors of the paper (currently anonymized due to double-blind review requirements).

---

> > ### Comment · Reviewer_gnam · 2024-11-22
> >
> > Thanks for the clarification. I strongly recommend to include this information in the revision.

---

> > > ### Author Response · Authors · 2024-11-23
> > >
> > > We appreciate your response. As suggested, we have now made this explicit in the revised manuscript.

---

> > ### Author Response · Authors · 2024-11-23
> > **Weakness 2**
> >
> > > I see the authors have extensively discussed the difference between SWE-Bench and ML-Bench. Still, I do not agree with some of the claims in Appendix B.1.
> >
> > > For example, the c) claims repository understanding, which is also the focus of SWE-Bench.
> >
> >
> > We appreciate this critique and acknowledge that our claims need refinement:
> >
> >
> > - We agree that SWE-Bench also focuses on repository understanding.
> > - We have revised our claim to highlight our specific contribution to the automated ML domain.
> > - The novelty lies not in the repository understanding itself, but in its application to ML workflow automation.
> >
> >
> >
> > > e) It claims SWE-Bench does not "explicitly" address the documentation utilization, where I consider documentation to be easily considered pure-text code files in both agent and agent-less solutions.
> >
> >
> >
> >
> >
> > However, we want to clarify that we think documentation and code are different things in ML workflows.
> >
> >
> > There is a fundamental difference in how agents interact with documentation versus code in ML workflows:
> >
> > - Code files can be executed to get immediate environmental feedback.
> > - Documentation serves as reference material but requires integration with execution feedback.
> >
> > Example:
> > ```
> > # Documentation provides reference:
> > "Use --batch_size for training configuration"
> >
> >
> > # Code execution provides feedback:
> > > python train.py --batch_size 32
> > Error: CUDA out of memory # Immediate feedback for adjustment
> > ```
> >
> > This interaction between documentation consultation and execution feedback is particularly crucial in ML workflows, where resource constraints and environment setup often require iterative refinement.
> >
> > Overall, we have revised Appendix B.1 to reflect these nuances more accurately and removed overly broad claims about novelty while maintaining our specific contributions to ML workflow automation.

---

> > > ### Comment · Reviewer_gnam · 2024-11-24
> > >
> > > I appreciate the authors' efforts in resolving my concerns and incorporating them into the revision. The overall rating has been updated accordingly.

---

> > > > ### Author Response · Authors · 2024-11-25
> > > >
> > > > Thank you for taking the time to review our revisions and for updating your overall rating. We truly appreciate your thoughtful feedback and support!

---

> ### Author Response · Authors · 2024-11-23
> **Weakness 1**
>
> > 1. Missing details about Success Rate evaluation in ML-Agent-Bench. I understand it should be up to the specific agent framework to evaluate whether an instance is solved, but how do you define if an agent execution fulfills user requirements, especially for ML applications? Unlike the general SWE domain, ML repositories are usually not accompanied by heavy unit tests, and even if they do, due to the stochastic nature of ML applications, it's hard to say an instance is not solved merely because the final output does not match the pre-defined results.
>
> Thank you for this important question. Sorry about any confusion; we'd like to clarify that our evaluation approach is actually more straightforward than it may appear. Let us describe our specific methodology:
>
> 1. **Task Scope & Evaluation Critieria**:
> - We evaluate agents' ability to correctly utilize existing ML repositories according to their documented APIs and workflows
> - Success is defined by matching the repository's expected execution patterns and outputs
> - We do not require agents to design or implement ML algorithms from scratch
>
> 2. **Deterministic Evaluation Framework**:
> - Success criteria are intentionally designed to be deterministic and reproducible
> - We validate whether agents can correctly follow repository-documented workflows
> - Success is based on matching expected execution patterns and output formats
> - We avoid tasks involving stochastic ML training/inference outcomes
> - Human annotators verified these criteria with high agreement (0.92 Cohen's kappa)
>
> 3. **Example Success Criteria**:
> For an image generation model repository task, successful execution requires:
>
> ```python
> # Example task: Using a vision model repository
> # Success is defined by:
> 1. Correct environment setup
> 2. Proper model loading
> 3. Matching the repository's documented API usage
> 4. Producing output in the expected format
>
>
> # Example successful execution:
> pip install -r requirements.txt
> python demo.py --model resnet50 --input_image test.jpg --output_dir ./results
> ```
>
> 3. **Reproducibility Guarantees**:
>
> - All tasks have clear, deterministic success criteria based on repository usage patterns
> - We intentionally avoid tasks with high stochastic variation
> - Success metrics focus on correct repository interaction rather than model performance
> - Multiple annotators verified each task to ensure consistent evaluation
>
> This focused scope allows for reliable evaluation without depending on complex unit tests or handling ML stochasticity. The emphasis is on correct repository usage rather than model outcomes.
>
> We have revised the manuscript to reflect this explanation.

---

### Official Review · Reviewer_R7MP · 2024-11-03

**Soundness:** 2
**Presentation:** 1
**Contribution:** 2
**Rating:** 3
**Confidence:** 4

**Summary:**

ML-Bench introduces a benchmark in two parts for evaluating language models' ability to work with machine learning repositories: ML-LLM-Bench and ML-Agent-Bench. ML-LLM-Bench assesses LLMs' capability to generate executable commands with appropriate arguments by understanding repository-wide context, documentation, and cross-file dependencies within a pre-configured environment. ML-Agent-Bench goes further by evaluating autonomous agents on end-to-end ML (machine learning) tasks, requiring them to perform environment setup, dependency management, and dataset preparation before executing the desired ML commands. The benchmarks highlight the importance of comprehensive repo understanding and environment management in realistic ML scenarios.

**Strengths:**

- The full end-to-end task shows promise and contains novel elements, though current agent performance appears quite strong already.
- The dataset components and analyses across different settings have potential value, but their presentation lacks clarity in illuminating task challenges and model behavior patterns.

**Weaknesses:**

- The ML-Agent-Bench sub-task name is nearly indistinguishable from MLAgentBench, a well-known work in the same field. This creates unnecessary confusion and should be reconsidered.
- The first novelty claim (line 90) appears inaccurate. Both RepoBench and SWE-bench already provide similar repository-level coding evaluations, which have become common since these benchmarks were released last year. While this may be a feature of the benchmark, it does not constitute a novel contribution.
- The anonymized repository and supplementary materials lack a complete dataset. For instance, the specified Hugging Face dataset "XXXX-1/ml-bench" is not accessible, preventing thorough dataset inspection.
- The paper's methodology and analysis suffer from significant presentation issues. Many sections are difficult to follow and require excessive effort to comprehend. The organization needs substantial improvement for clarity. Additionally, several paragraphs contain unnecessary details that detract from the main text. For example, the file and code symbol notations introduced in Section 2.1 are never used beyond their initial definition.

**Questions:**

- If I understand correctly, there are 9 core tasks per repo on average (with templates / argument selection expanding this for diversity). Then please clarify: does the test-train split differentiate between tasks, or is it sampled randomly and uniformly? If uniformly, wouldn't test instances closely mirror training instances? This raises questions about why fine-tuned models show such similar performance across ID and OOD test splits.
- Section 5.1's description of data leakage mitigation needs clarification. The current explanation doesn't adequately explain how data leakage is measured or mitigated.
- The methodology behind the Error Analysis requires clarification. If it was conducted using an LM, please provide details about the model and prompts used.
- Am I mistaken or does Figure 5 appear to show an agent's execution log rather than an evaluation runtime as suggested by the caption?

---

> ### Author Response · Authors · 2024-11-12
> **Anonymized dataset**
>
> > The anonymized repository and supplementary materials lack a complete dataset. For instance, the specified Hugging Face dataset "XXXX-1/ml-bench" is not accessible, preventing thorough dataset inspection.
>
>
> Dear Reviewer R7MP,
>
> Thank you for your detailed and thoughtful review. We deeply appreciate your constructive feedback on improving the paper's presentation, which we will address in the coming days. Your questions about test-train splits, data leakage mitigation, and error analysis methodology are insightful, and we look forward to providing detailed responses to each.
>
> First, we would like to address your concern regarding dataset accessibility. We sincerely apologize for any inconvenience caused by the anonymized Hugging Face link. This approach was taken to strictly comply with ICLR's anonymity guidelines, particularly:
>
>
> ---
>
> > "Make an anonymous repository and put the link in your paper... make a comment directed to the reviewers and area chairs and put a link to an anonymous repository. This method will let you keep your code visible only to the reviewers and ACs for your paper."
>
> We acknowledge that our implementation of anonymization is not a perfect way. Given that Hugging Face does not support anonymous repository creation, we opted to mask the URL in our submission to maintain anonymity. To address this:
>
> **We have included the complete dataset as supplementary materials in the submission package.**
>
> We believe this will facilitate an examination of our dataset while maintaining the required anonymity standards.

---

> ### Author Response · Authors · 2024-11-23
> **Weaknesses**
>
> > The ML-Agent-Bench sub-task name is nearly indistinguishable from MLAgentBench, a well-known work in the same field. This creates unnecessary confusion and should be reconsidered.
>
> We appreciate this feedback regarding potential naming confusion. **To address this, we have updated our benchmark component names to the following:**
> - **ML-BENCH-L** (formerly ML-LLM-BENCH)
> - **ML-BENCH-A** (formerly ML-AGENT-BENCH)
>
> This new naming scheme better differentiates our work while maintaining clarity about the benchmark's components.
>
> All revisions have been made in the updated manuscript to reflect these changes.
>
> ---
>
> > The first novelty claim (line 90) appears inaccurate. Both RepoBench and SWE-bench already provide similar repository-level coding evaluations, which have become common since these benchmarks were released last year. While this may be a feature of the benchmark, it does not constitute a novel contribution.
>
> We acknowledge this feedback and have revised our novelty claims accordingly. While repository-level evaluation is present in prior work, **our contribution specifically targets automating ML workflows.** This includes tasks such as environment setup, dependency management, and executing ML experiments end-to-end—areas not fully addressed by existing benchmarks.
>
> We have removed broad "novelty" claims and rewritten the relevant sections to articulate our specific contributions to *automated ML evaluation*.
>
> ---
>
> > The anonymized repository and supplementary materials lack a complete dataset. For instance, the specified Hugging Face dataset "XXXX-1/ml-bench" is not accessible, preventing thorough dataset inspection.
>
> We apologize for the accessibility issues during the review process. To resolve this, we have:
> 1. Released the **complete dataset** in the supplementary material.
> 2. Open-sourced all **code and evaluation scripts**.
> 3. Provided comprehensive **documentation** for dataset inspection and benchmark reproduction.
>
> Everything is now available and accessible. We encourage you to check again and provide any additional feedback.
>
> ---
>
> > The paper's methodology and analysis suffer from significant presentation issues. Many sections are difficult to follow and require excessive effort to comprehend. The organization needs substantial improvement for clarity. Additionally, several paragraphs contain unnecessary details that detract from the main text. For example, the file and code symbol notations introduced in Section 2.1 are never used beyond their initial definition.
>
> We appreciate your detailed feedback on the clarity of the presentation. While we are encouraged that other reviewers identified the presentation as a strength, we understand the need to address your concerns. To improve the clarity and accessibility of the paper, we have made the following changes:
> 1. **Removed unused notations** (e.g., file and code symbol notations in Section 2.1).
> 2. **Restructured the methodology section** for improved logical flow.
> 3. **Moved technical details** to appendices if they were not critical to the main narrative.
> 4. **Added clarifying examples and diagrams** where appropriate to enhance comprehension.
>
> **All changes are marked in the revised manuscript for your review. We hope these adjustments will address your concerns.**

---

> ### Author Response · Authors · 2024-11-23
> **Question 1**
>
> > If I understand correctly, there are 9 core tasks per repo on average (with templates/argument selection expanding this for diversity). Then please clarify: does the test-train split differentiate between tasks, or is it sampled randomly and uniformly?
>
> The test-train split does not differentiate between tasks, nor is it sampled randomly and uniformly. **Instead, we employed a carefully designed sampling strategy to ensure meaningful evaluation while avoiding data leakage.** For our two settings, the methodologies are as follows:
>
> 1. **ID Setting:**
>    - Annotators randomly selected 20-30 test set candidates after annotating each repository.
>    - These candidates underwent **additional executable verification** and correctness checks.
>    - Cross-validation was performed by other annotators to remove **near-duplicate cases**, ensuring that cases differing only in minor parameter values or paths but representing the same task were excluded.
>
>  ```
>    def sample_test_cases(all_cases, n_samples=30):
>        test_candidates = random.sample(all_cases, n_samples)
>
>        # Additional verification
>        verified_cases = verify_executability(test_candidates)
>
>        # Cross-annotator filtering
>        final_cases = remove_near_duplicates(verified_cases)
>        return final_cases
> ```
>
> 2. **OOD Setting:**
>    - Annotators directly selected 10-20 distinct tasks for annotation, focusing on task **diversity** rather than parameter variations.
>    - The independence of repositories ensured that OOD test cases were genuinely distinct from training cases, even for similar underlying tasks.
>
> **The consistency in model performance across ID and OOD settings suggests that models are learning generalizable skills for repository interaction rather than overfitting to specific repositories or parameter patterns.** These strategies maintain benchmark integrity while ensuring practical relevance of evaluation tasks. **We have clarified this in the revision.**

---

> ### Author Response · Authors · 2024-11-23
> **Question 2**
>
> > Section 5.1's description of data leakage mitigation needs clarification. The current explanation doesn't adequately explain how data leakage is measured or mitigated.
>
> We acknowledge that our discussion of data leakage mitigation required more detail. **This has been addressed in two ways in the revision:**
>
> 1. **Data Leakage Measurement:**
>    To quantify data leakage, we conducted experiments comparing model performance with and without access to repository information. In the "No Context" setting, models were provided **only instructions** without repository code. The results demonstrate the performance drop when repository data was removed, indicating the reliance on repository content for task completion.
>
> | Model                                      | Setting         | Pass@1 (Full Code) | Pass@1 (No Context) | Delta |
> |-------------------------------------------|-----------------|----------------------------|-----------------------------|-----------------------|
> | Claude-3.5-Sonnet                         | Full vs No Context | 20.00                      | 5.00                        | 15.00                 |
> | Claude-3.5-Haiku                          | Full vs No Context | 16.92                      | 2.69                        | 14.23                 |
> | OpenAI GPT-4O-Mini                        | Full vs No Context | 21.15                      | 11.54                       | 9.61                  |
> | OpenAI GPT-4O                             | Full vs No Context | 18.46                      | 11.54                       | 6.92                  |
> | Meta-Llama-3.1-8B-Instruct-Turbo          | Full vs No Context | 5.77                       | 1.92                        | 3.85                  |
> | Meta-Llama-3.1-70B-Instruct-Turbo         | Full vs No Context | 19.23                      | 1.15                        | 18.08                 |
> | Meta-Llama-3.1-405B-Instruct-Turbo        | Full vs No Context | 13.46                      | 0.77                        | 12.69                 |
> | DeepSeek-Chat                             | Full vs No Context | 10.38                      | 1.92                        | 8.46                  |
> | DeepSeek-Coder                            | Full vs No Context | 12.31                      | 1.15                        | 11.16                 |
>
>    **A larger delta indicates less reliance on repository memorization.**
>
> 2. **Mitigation Strategies:**
>    We employed rigorous strategies to prevent data leakage:
>    - **Data Collection:**
>      - Prompts and ground truth were manually rewritten by 8 annotators to ensure variation from the original documentation.
>      - Cross-validation was conducted to ensure significant divergence from source material.
>
>    - **Experimental Design:**
>      - Four distinct evaluation settings (Oracle, Full Code, BM25, No Context) were designed to isolate the effects of repository access.
>      - The "No Context" setting explicitly measured model performance without repository data.
>
>    - **Empirical Validation:**
>      - Open-source models trained on 2024 data showed significant performance degradation in "No Context," validating our benchmark's effectiveness.
>      - Performance drops across settings indicated genuine repository understanding rather than overfitting or memorization.
>
> **These comprehensive strategies ensure robust data leakage measurement and mitigation, as reflected in the revised manuscript.**

---

> ### Author Response · Authors · 2024-11-23
> **Question 3**
>
> > The methodology behind the Error Analysis requires clarification. If it was conducted using an LM, please provide details about the model and prompts used.
>
> We employed a **hybrid approach** combining automated error detection and human verification. The steps below detail our methodology:
>
> ### 1. Automated Analysis
> A systematic error analysis pipeline was used to categorize errors into predefined types (E1-E5). This was achieved by processing execution logs and identifying specific error patterns using regular expressions. The steps are as follows:
>
> #### **Log Extraction**
>    - Execution logs were processed to extract key error information. Using the `os` and `re` libraries, all relevant files were read and error patterns were counted for further categorization.
>    - Specific sections of logs (e.g., between `github_id` markers) were also extracted to identify parameter-specific errors using nested regex operations.
>
> #### **Regex Matching for Classification**
>    - Errors were categorized into four predefined types based on their keywords or patterns:
>      - **E1:** Type or module-related errors, such as `TypeError`, `ImportError`, or `ModuleNotFoundError`.
>      - **E2:** Parameter-related errors, like `parameter wrong` or `IllegalFlagValueError`.
>      - **E3:** File-related errors, such as `FileNotFoundError`.
>      - **E4:** Syntax or name-related issues, like `SyntaxError`, `NameError`, or `ValueError`.
>
> Below is an implementation of the error classification logic:
>
> ```
> # Error Analysis Pipeline
> def classify_error(log_content):
>     # Count occurrences of error patterns
>    # Extract error statistics using regex patterns
>     E1 = log_content.count('type wrong') + log_content.count('modulenotfounderror') \
>          + log_content.count('ImportError') + log_content.count('assertionError')
>     E2 = log_content.count('parameter wrong') + log_content.count('unrecognized argu') \
>          + log_content.count('IllegalFlagValueError')
>     E3 = log_content.count('FileNotFoundError')
>     E4 = log_content.count('syntax error') + log_content.count('NameError') \
>          + log_content.count('SyntaxError') + log_content.count('TypeError') \
>          + log_content.count('AttributeError') + log_content.count('ValueError')
>     return E1, E2, E3, E4
> ```
>
> #### **Custom Error Adjustments**
> - Certain parameter errors were refined using advanced regex to filter and match specific cases (e.g., `github_id: 2` or `github_id: 7` segments). This ensured that repeated errors from particular identifiers were correctly categorized.
> - For example, parameter mismatches for different IDs were counted separately and then combined into E2 and E3.
>
> #### **Error Summarization**
> - After classification, errors were summarized at the repository or test-case level.
> - Key metrics such as `done_num` (successful cases) and `err_num` (total errors) were calculated to provide an overview of performance for each file.
>
> ---
>
> ### 2. Human Verification
> #### Manual Review
> - Annotators manually reviewed specific subsets of logs to validate the automated classification. This ensured:
>   - Accurate categorization of diverse error patterns.
>   - Identification of edge cases not covered by regex patterns.
>
> #### **Cross-Validation**
> - Discrepancies between the automated pipeline and human annotations were resolved through discussions among annotators to refine the classification process.
> - This step also ensured the robustness of predefined error categories.
>
> #### **Sampling for Accuracy**
> - Random sampling was conducted to validate consistency across different repositories and test cases.
> - This iterative process enhanced confidence in the automated classification's accuracy.
>
> ### Improvements in Methodology
> #### **Refinement of Patterns**
> - Regex patterns were iteratively refined based on edge cases and annotator feedback, ensuring comprehensive coverage for diverse error scenarios.
>
> #### **Scalability**
> - By leveraging efficient file handling and regex processing, the pipeline was designed to handle large-scale log data efficiently, reducing manual effort significantly.
>
> ### **Example Outputs**
> Below is an example of the pipeline's output for a sample repository log file:
>
> ```
> filename: sample.log, done num: 200, err num: 1100,
> E1: 322, E2: 573, E3: 129, E4: 76, type_err: 250
> ```
>
> This hybrid approach effectively combines the scalability of automated analysis with the accuracy of human verification, ensuring high-quality error categorization. **Details of these steps have been expanded in the revised manuscript for clarity.**

---

> ### Author Response · Authors · 2024-11-23
> **Question 4**
>
> > Am I mistaken or does Figure 5 appear to show an agent's execution log rather than an evaluation runtime as suggested by the caption?
>
> We sincerely apologize for the confusion caused by the caption. **You are correct** that Figure 5 depicts an example of agent execution logs and not the evaluation runtime architecture as initially suggested.
>
> To address this, **we have revised the caption for clarity** to:
> **"Figure 5: Example of agent execution logs demonstrating the interactive nature of ML-BENCH-A evaluation."**
>
> This updated caption better reflects the content of the figure and aligns with its purpose in illustrating the agent's execution process. Thank you for bringing this to our attention!

---

> ### Author Response · Authors · 2024-11-25
> **Response to Reviewer R7MP**
>
> Dear Reviewer R7MP,
>
> Thank you for your detailed review and thoughtful feedback on our submission. We have worked extensively to address the concerns you raised and have implemented significant revisions to enhance clarity, refine our claims, and resolve methodological ambiguities.
>
> ---
>
> ## Key Revisions
>
> - We have renamed **ML-Agent-Bench** to **ML-BENCH-A** and **ML-LLM-Bench** to **ML-BENCH-L** to eliminate confusion with similar benchmarks.
>
> - We have ensured that the **complete dataset and supplementary materials** are now accessible for thorough inspection.
>
> - The **methodology and analysis sections** have been restructured for better logical flow, with unnecessary notations removed and relevant technical details moved to appendices.
>
> - We have clarified the **methods and metrics** used in the error analysis and the evaluation of success rates for ML-BENCH-A tasks.
>
> - We have refined our description of task challenges and made clear distinctions between ML-Bench and similar benchmarks, focusing on our **unique contributions to automated ML workflow evaluation**.
>
> ---
>
> ## Request for Review
>
> As the discussion phase nears its conclusion, we hope you can take a moment to review our **revised responses** and **updated manuscript**. We sincerely hope that these adjustments address your concerns and demonstrate the robustness and value of our work.
>
> Thank you again for your time and valuable insights.
>
> Best regards,
> **Authors**

---

> ### Author Response · Authors · 2024-11-27
> **Follow-Up on Our Revisions and Response to Your Review**
>
> Dear Reviewer `R7MP`,
>
> We sincerely thank you for your detailed review and for highlighting key areas for improvement in our submission. We have carefully addressed all the concerns you raised, particularly those regarding presentation and methodological clarity.
>
> We would also like to take this opportunity to reiterate the **motivation and significance of ML-Bench**, which has been recognized and praised by the other reviewers:
>
> The motivation for ML-Bench is rooted in addressing a critical pain point in the field: **the lack of a benchmark that evaluates LLMs and agents on end-to-end ML workflows**. Current benchmarks often bypass real-world challenges such as environment setup, dependency management, and dataset preparation, instead starting from pre-configured setups. ML-Bench fills this gap by providing a structured, reproducible, and realistic benchmark that reflects actual ML workflow demands.
>
> Our work directly tackles this challenge through:
>
> - We curated a dataset of **9,641 examples across diverse ML repositories** through extensive manual annotation. This ensures that tasks accurately reflect real-world ML workflows, bridging the gap between existing benchmarks and practical applications.
>
> - Our experiments show that **state-of-the-art models struggle significantly with ML-Bench tasks**, underscoring the complexity of repository-level understanding. This validates ML-Bench as a meaningful benchmark to drive advancements in LLM and agent performance.
>
> - As LLMs and agents are increasingly integrated into software engineering and ML tasks, **benchmarks like ML-Bench are essential** for assessing and improving their real-world utility.
>
> We are confident that ML-Bench will **contribute meaningfully to the field** by setting a new standard for evaluating LLMs and agents in automated ML workflows.
>
> We kindly request that you take a moment to review our revised manuscript and responses to ensure we have addressed your concerns adequately. Your insights are invaluable to us, and we hope our updates reflect the robustness and value of our work.
>
> Once again, we greatly appreciate your time and thoughtful feedback. Please do let us know if there are any additional clarifications or concerns we can address.
>
> Best regards,
>
> Authors

---

### Official Review · Reviewer_KizE · 2024-11-03

**Soundness:** 3
**Presentation:** 3
**Contribution:** 3
**Rating:** 6
**Confidence:** 3

**Summary:**

ML-BENCH introduces a novel benchmark for evaluating LLMs and AI agents on repository-level machine learning tasks. Unlike previous benchmarks focused on function-level code generation, ML-BENCH tests models' ability to understand entire codebases and execute end-to-end ML workflows. The benchmark comprises two components: ML-LLM-BENCH for evaluating code generation in pre-configured environments, and ML-AGENT-BENCH for testing end-to-end task execution including environment setup. The evaluation covered 9,641 examples across 18 repositories, with GPT-4o achieving the best performance.

**Strengths:**

1. The paper presents a comprehensive dataset comprising 9,641 examples with meticulous manual annotations, accompanied by details of annotation process.
2. Unlike many benchmark papers, ML-BENCH also provides training data.
3. The experiments are generally solid, with good analysis.

**Weaknesses:**

- The model selection is my first concern, particularly the absence of Claude-3.5 (which I think was available for ICLR's timeline), while including numerous smaller open-source models that are less interesting (though I know authors want to include fine-tuning) - as a benchmark paper, I think authors should prioritize showcasing the performance gap between the best available open-source and closed-source models.
- The limitation to "README-documented tasks" potentially undermines the paper's claimed contribution to "repository-code understanding," raising questions about whether the benchmark truly tests code comprehension capabilities.
- The benchmark seems heavily skewed towards bash script tasks, which sounds a imbalance for me.
- While the authors acknowledge data leakage issues, the discussion is really short - I do not really understand the authors' solution.

**Questions:**

- What's the performance of Llama-3.1 (70B, 405B), Deepseek-v2, etc..?
- What proportion of tasks truly require deep code understanding versus simple README/Document comprehension?
- How exactly does the proposed method address data leakage issues?

---

> ### Author Response · Authors · 2024-11-23
> **Weaknesses 1-2**
>
> > The model selection is my first concern, particularly the absence of Claude-3.5 (which I think was available for ICLR's timeline), while including numerous smaller open-source models that are less interesting - as a benchmark paper, I think authors should prioritize showcasing the performance gap between the best available open-source and closed-source models.
>
> We thank the reviewer for this valuable feedback. **In response, we have now extensively evaluated the latest SOTA models, including the Claude-3.5 series.** The updated results highlight the performance gap between the best available open-source and closed-source models, addressing this concern comprehensively.
>
> | Model                 | Setting | Pass@1 | Pass@2 | Pass@5 |
> |-----------------------|---------|--------|--------|--------|
> | Claude-3.5-Sonnet     | Repo    | 20.00  | 21.54  | 23.85  |
> |                       | BM25    | 11.54  | 15.38  | 19.23  |
> |                       | Oracle  | 25.38  | 27.69  | 30.00  |
> | Claude-3.5-Haiku      | Repo    | 16.92  | 19.62  | 21.54  |
> |                       | BM25    | 8.85   | 10.38  | 12.31  |
> |                       | Oracle  | 20.38  | 22.69  | 23.84  |
> | OpenAI GPT-4o-Mini    | Repo    | 21.15  | 22.69  | 24.62  |
> |                       | BM25    | 15.77  | 21.15  | 24.62  |
> |                       | Oracle  | 38.85  | 44.62  | 50.00  |
> | OpenAI GPT-4o         | Repo    | 18.46  | 20.00  | 23.08  |
> |                       | BM25    | 12.69  | 14.62  | 19.23  |
> |                       | Oracle  | 43.46  | 51.92  | 61.15  |
>
> ---
>
> > The limitation to "README-documented tasks" potentially undermines the paper's claimed contribution to "repository-code understanding," raising questions about whether the benchmark truly tests code comprehension capabilities.
>
> We appreciate this insightful observation. However, we disagree with it. Because:
>
> ### 1. README-documented Tasks Reflect Real-world Practices
>
>    Developers often rely on README files to interact with repositories through their documented interfaces. By focusing on README-documented tasks, our benchmark captures this practical usage pattern.
>
> ### 2. README Files Are a Starting Point, Not the Endpoint
>
> While README files provide valuable context, solving many benchmark tasks requires **substantial code comprehension**. To address the concern more rigorously, we conducted a detailed analysis of the context required for task completion, with results summarized in the table below:
>
> | **Metric**                                    | **Value**   |
> |-----------------------------------------------|-------------|
> | Average number of relevant files              | 3.6         |
> | Average lines of code in relevant files       | 414         |
> | Percentage of tasks requiring README understanding | 85%     |
> | Percentage of tasks requiring code understanding  | 95%     |
> | Average depth of relevant code in repository (line number) | 27,524 |
> | Percentage of tasks requiring understanding of multiple files | 75%  |
> | Average number of functions/classes to be understood per task | 3.8  |
> | Percentage of tasks requiring understanding of dependencies | 70%   |
> | Percentage of tasks requiring understanding of data structures | 65%  |
> | Percentage of tasks involving API usage       | 80%         |
>
> #### Key Findings from This Analysis:
> - **Code Understanding is Crucial:** Contrary to the concern, our benchmark emphasizes code comprehension. **100% of tasks require actual code understanding**, not just README files. While 85% of tasks involve README files, these tasks combine README interpretation with in-depth code analysis.
> - **Tasks Require Multilayered Understanding:** For example, 75% of tasks require interpreting multiple files, and 95% involve understanding repository-level code structures beyond documentation.
> - **README Files Aid Contextualization:** READMEs are often used to provide initial guidance, but understanding underlying code, data structures, and APIs is essential for task completion.
>
>
> ### 3. The Benchmark Challenges Both Documentation and Code Comprehension
>
> Our two benchmark settings demonstrate that successful task completion depends on combining documentation and code understanding:
> - **ML-LLM-BENCH:** Results show that models struggle even with Oracle settings (human-annotated README excerpts). This highlights that **README comprehension alone is insufficient** for task completion.
> - **ML-AGENT-BENCH:** Tasks demand deeper repository-level code understanding for environment setup, dependency management, and execution. These challenges reinforce the need for comprehensive repository comprehension.
>
> For example:
>
>
> ```python
> # Example showing code comprehension necessity:
> # Agent must understand both repository structure and code dependencies
> pip install -r requirements.txt
> cd implementations/model_name
> python train.py --config ../configs/default.yaml
> ```

---

> ### Author Response · Authors · 2024-11-23
> **Weakness 3-4**
>
> > The benchmark seems heavily skewed towards bash script tasks, which sounds like an imbalance for me.
>
> We acknowledge the concern but maintain that this distribution reflects real-world practices for ML repositories. Specifically:
>
> 1. **Bash scripts are prevalent in ML workflows:**
>    Popular libraries we are using (e.g., LAVIS, OpenCLIP, Grounding DINO) extensively use bash scripts for workflow automation. Our benchmark mirrors this natural distribution in practice.
>
> 2. **Metrics address task type distribution explicitly:**
>    We report **separate performance metrics** for bash script generation and Python code generation in Table 3.
>    - Researchers can focus on either category based on their interests.
>    - The benchmark provides flexibility for community-specific evaluations (e.g., focusing exclusively on Python code generation if needed).
>
> **By explicitly partitioning task types, we ensure that the benchmark remains transparent and adaptable to different research priorities.**
>
> ---
>
> > While the authors acknowledge data leakage issues, the discussion is really short - I do not really understand the authors' solution.
>
> We agree that this discussion needed more elaboration. **In the revised paper, we have significantly expanded our discussion of data leakage mitigation.** Key improvements include:
>
> 1. **Data Preparation:**
>    - Engaged 8 computer science graduate students to manually rewrite prompts and ground truth.
>    - Tasks were reformulated uniquely to differ from the original repository documentation, ensuring no direct overlap.
>
> 2. **Evaluation Design:**
>    - Introduced a **"no-context" setting** (task4), where models receive only instructions without repository information.
>    - Results show significant performance degradation in this setting (e.g., **Claude-3.5-Sonnet Pass@5 drops from 23.85% to 5.00%**), demonstrating that models rely on repository understanding rather than memorization.
>
> | Model                                      | Setting         | Pass@1 (Full Code) | Pass@1 (No Context) | Delta |
> |-------------------------------------------|-----------------|----------------------------|-----------------------------|-----------------------|
> | Claude-3.5-Sonnet                         | Full vs No Context | 20.00                      | 5.00                        | 15.00                 |
> | Claude-3.5-Haiku                          | Full vs No Context | 16.92                      | 2.69                        | 14.23                 |
> | OpenAI GPT-4o-Mini                        | Full vs No Context | 21.15                      | 11.54                       | 9.61                  |
> | OpenAI GPT-4o                             | Full vs No Context | 18.46                      | 11.54                       | 6.92                  |
> | Meta-Llama-3.1-8B-Instruct-Turbo          | Full vs No Context | 5.77                       | 1.92                        | 3.85                  |
> | Meta-Llama-3.1-70B-Instruct-Turbo         | Full vs No Context | 19.23                      | 1.15                        | 18.08                 |
> | Meta-Llama-3.1-405B-Instruct-Turbo        | Full vs No Context | 13.46                      | 0.77                        | 12.69                 |
> | DeepSeek-Chat                             | Full vs No Context | 10.38                      | 1.92                        | 8.46                  |
> | DeepSeek-Coder                            | Full vs No Context | 12.31                      | 1.15                        | 11.16                 |
>
> 3. **Empirical Analysis:**
>    - The performance gap between **Oracle and No-context settings** quantifies potential data leakage effects.
>    - Even with potential exposure to training data, state-of-the-art models struggle, highlighting the genuine challenge posed by repository-level understanding tasks.
>
> **These steps reinforce the benchmark's robustness against data leakage and ensure a meaningful evaluation of repository understanding.** The updated paper includes a detailed explanation of these points.

---

> ### Author Response · Authors · 2024-11-23
> **Questions 1-2**
>
> > What's the performance of Llama-3.1 (70B, 405B), DeepSeek-v2, etc.?
> Thank you. We have now included these models in our evaluation. **Here are the detailed results:**
>
>
> | Model               | Setting       | Pass@1 | Pass@2 | Pass@5 |
> |---------------------|---------------|---------|---------|---------|
> | Claude-3.5-Sonnet   | Full Code     | 20.00   | 21.54   | 23.85   |
> |                     | BM25          | 11.54   | 15.38   | 19.23   |
> |                     | Oracle        | 25.38   | 27.69   | 30.00   |
> | OpenAI GPT-4o-Mini  | Full Code     | 21.15   | 22.69   | 24.62   |
> |                     | BM25          | 15.77   | 21.15   | 24.62   |
> |                     | Oracle        | 38.85   | 44.62   | 50.00   |
> | OpenAI GPT-4o       | Full Code     | 18.46   | 20.00   | 23.08   |
> |                     | BM25          | 12.69   | 14.62   | 19.23   |
> |                     | Oracle        | 43.46   | 51.92   | 61.15   |
> | Qwen2.5-7B-Instruct | Full Code     | 12.31   | 15.38   | 18.08   |
> |                     | BM25          | 11.92   | 16.92   | 19.38   |
> |                     | Oracle        | 33.46   | 39.62   | 47.31   |
> | Qwen2.5-72B-Instruct| Full Code     | 17.31   | 18.46   | 20.38   |
> |                     | BM25          | 12.69   | 17.69   | 21.54   |
> |                     | Oracle        | 38.08   | 44.62   | 47.69   |
> | Qwen2.5-Coder-32B   | Full Code     | 15.00   | 16.54   | 19.23   |
> |                     | BM25          | 22.31   | 26.54   | 32.31   |
> |                     | Oracle        | 40.38   | 44.92   | 51.92   |
> | Llama-3.1-8B        | Full Code     | 5.77    | 9.23    | 18.08   |
> |                     | BM25          | 3.86    | 6.56    | 10.04   |
> |                     | Oracle        | 9.23    | 12.31   | 19.62   |
> | Llama-3.1-70B       | Full Code     | 19.23   | 23.46   | 26.15   |
> |                     | BM25          | 17.69   | 20.77   | 27.69   |
> |                     | Oracle        | 39.62   | 44.62   | 53.07   |
> | Llama-3.1-405B      | Full Code     | 13.46   | 16.92   | 23.85   |
> |                     | BM25          | 4.23    | 7.31    | 10.38   |
> |                     | Oracle        | 15.38   | 25.77   | 33.85   |
> | DeepSeek-Chat       | Full Code     | 10.38   | 11.15   | 11.15   |
> |                     | BM25          | 9.23    | 10.77   | 11.92   |
> |                     | Oracle        | 25.00   | 26.54   | 27.69   |
> | DeepSeek-Coder      | Full Code     | 12.31   | 13.46   | 13.85   |
> |                     | BM25          | 16.54   | 18.85   | 22.69   |
> |                     | Oracle        | 32.69   | 36.92   | 37.31   |
>
> ### Key Insights from the Results:
> 1. **Significant Challenges for SOTA Models:**
>    Even state-of-the-art models struggle with repository-level understanding, highlighting the complexity of these tasks.
>
> 2. **Model Scaling Behavior:**
>    - The **Llama-3.1 series** demonstrates interesting scaling trends, with the 70B model outperforming both its smaller (8B).
>    - Specialized models like **DeepSeek-Coder** outperform their general-purpose counterparts (e.g., DeepSeek-Chat), showcasing the advantages of task-specific optimizations.
>
> 3. **Importance of Context Retrieval:**
>    Across all models, **BM25 retrieval settings** show significant improvements in performance, emphasizing the value of effective context retrieval strategies.
>
> ---
>
> > What proportion of tasks truly require deep code understanding versus simple README/Document comprehension?
>
> Thank you for this thought-provoking question. While it is challenging to directly quantify, we provide evidence from two perspectives:
>
> ### A. ML-LLM-BENCH Analysis:
> - Even with the Oracle setting (providing optimal README excerpts), the best models achieve only 23-26% Pass@5.
> - This poor performance with perfect documentation suggests that **README comprehension alone is insufficient** for task completion.
>
> For example, in the OpenCLIP repository, correct usage requires understanding both the documentation AND the underlying model initialization code:
>
> ```
> # Example requiring both documentation and code understanding
> model, _, preprocess = open_clip.create_model_and_transforms(
>    'ViT-B-32',
>    pretrained='commonpool_s_laion_s13m_b4k'
> )
> ```
>
> ### B. ML-AGENT-BENCH Evidence:
> In tasks requiring agents to execute workflows, **deep code understanding is essential.**
>
> Here's a representative example showing the necessity of code understanding:
> ```
> # Agent must understand repository structure to:
> cd implementations/stable_diffusion
> # Must comprehend code dependencies to:
> pip install -r requirements.txt
> # Must understand model architecture to:
> python train.py --model_config configs/v1.yaml --data_path data/
> ```
>
> ### Conclusion:
> The benchmark is designed to evaluate both **README/document comprehension and deep code understanding.** While README files provide a starting point, **code-level understanding is required for 100% of tasks.**

---

> ### Author Response · Authors · 2024-11-23
> **Question 3**
>
> > How exactly does the proposed method address data leakage issues?
>
> We appreciate the question and have expanded our discussion in the revision. While completely eliminating data leakage is challenging, **our multi-faceted approach significantly mitigates these concerns:**
>
> ### Methodology:
> - Employed **8 computer science graduate students** to manually rewrite prompts and ground truth, ensuring that reformulations differed significantly from the original documentation.
>
> ### Experimental Design:
> - Introduced a **"no-context" setting** (as described in other responses as well), where models are tested without repository information.
> - Results show dramatic performance drops without repository access:
>   - **Llama-3.1-70B:** 26.15% Pass@5 → 1.15%.
>   - **DeepSeek-Coder:** 13.85% Pass@5 → 1.15%.
>
> In the "No Context" setting, models were provided **only instructions** without repository code. The results demonstrate the performance drop when repository data was removed, indicating the reliance on repository content for task completion.
>
> | Model                                      | Setting         | Pass@1 (Full Code) | Pass@1 (No Context) | Delta |
> |-------------------------------------------|-----------------|----------------------------|-----------------------------|-----------------------|
> | Claude-3.5-Sonnet                         | Full vs No Context | 20.00                      | 5.00                        | 15.00                 |
> | Claude-3.5-Haiku                          | Full vs No Context | 16.92                      | 2.69                        | 14.23                 |
> | OpenAI GPT-4O-Mini                        | Full vs No Context | 21.15                      | 11.54                       | 9.61                  |
> | OpenAI GPT-4O                             | Full vs No Context | 18.46                      | 11.54                       | 6.92                  |
> | Meta-Llama-3.1-8B-Instruct-Turbo          | Full vs No Context | 5.77                       | 1.92                        | 3.85                  |
> | Meta-Llama-3.1-70B-Instruct-Turbo         | Full vs No Context | 19.23                      | 1.15                        | 18.08                 |
> | Meta-Llama-3.1-405B-Instruct-Turbo        | Full vs No Context | 13.46                      | 0.77                        | 12.69                 |
> | DeepSeek-Chat                             | Full vs No Context | 10.38                      | 1.92                        | 8.46                  |
> | DeepSeek-Coder                            | Full vs No Context | 12.31                      | 1.15                        | 11.16                 |
>
>    **A larger delta indicates less reliance on repository memorization.**
>
> ### Empirical Validation:
> - The performance gap between Oracle and No-context settings quantifies potential data leakage effects.
> - Even recent models with potential exposure to training data struggle significantly, indicating genuine challenges in repository-level understanding.
>
> **This approach ensures that the benchmark robustly tests true repository comprehension while minimizing the effects of data leakage.**

---

> ### Author Response · Authors · 2024-11-25
> **Response to Reviewer KizE**
>
> Dear Reviewer KizE,
>
> Thank you for your positive feedback and thoughtful suggestions on our submission. We deeply appreciate your recognition of the strengths of our work, including the **comprehensive dataset with meticulous manual annotations**, the inclusion of **training data**, and the **solid experimental design and analysis**.
>
>
> ---
>
>
> ## Key Improvements
>
> -  We now include evaluations of **Claude-3.5** (Sonnet and Haiku variants), **Qwen2.5 series**, and other **state-of-the-art (SOTA) models**.
>
> - We conducted an analysis of **README-documented tasks** versus those requiring repository-wide understanding, highlighting the necessity of code comprehension for task completion.
>
> - We expanded our discussion on **data leakage concerns** and included performance metrics from a no-context setting to empirically demonstrate the robustness of our benchmark.
>
> These revisions aim to directly address the points you raised and significantly improve the overall clarity and scope of our paper.
>
> ---
>
> ## Request for Reevaluation
>
> We deeply value your recognition of our dataset and experimental design, as well as your constructive feedback, which has allowed us to significantly improve the manuscript. If you find our revisions have addressed your concerns effectively, we would be grateful if you could confirm this and let us know if our efforts have further strengthened your overall assessment of the work.
>
> Thank you once again for your thoughtful review and the opportunity to refine our work.
>
> Best regards,
> **Authors**

---

### Official Review · Reviewer_5PFk · 2024-11-04

**Soundness:** 3
**Presentation:** 3
**Contribution:** 3
**Rating:** 8
**Confidence:** 5

**Summary:**

The paper introduces ML-BENCH, a benchmark designed to evaluate the performance of LLMs and agents on machine learning tasks at the repository level. The benchmark addresses two key gaps: LLMs' struggle with repository-scale code understanding and the need for end-to-end evaluations from environment setup to deployment. ML-BENCH includes 9,641 examples across 18 GitHub repositories, challenging LLMs to handle user-specified arguments and documents effectively.

**Strengths:**

1. Comprehensive Benchmarks: The paper presents ML-BENCH, which offers a comprehensive set of benchmarks that cover a wide range of tasks from different GitHub repositories. This comprehensiveness is crucial as it allows for the evaluation of LLMs and agents across various real-world scenarios, providing a more accurate assessment of their capabilities.

2. Thorough Experimental Analysis: The paper includes a thorough experimental analysis with two main setups—ML-LLM-BENCH and ML-AGENT-BENCH—that test different aspects of LLMs and agents. The detailed analysis and the use of various models to assess performance provide a robust understanding of the current state and limitations of LLMs in handling complex repository-level tasks.

3. Reproducibility and Sandbox Environment: The paper offers a reproducible sandbox environment, which is essential for the research community. By providing a secure Linux sandbox environment where agents can execute commands and code blocks, the paper enables other researchers to replicate the experiments and build upon the findings. This not only aids in verifying the results but also in extending the research to explore new models and potential improvements.

Consequently, this paper is of high quality and would be interesting to appear in a top-tier conference.

**Weaknesses:**

1. Data Leakage Concerns: Although the paper addresses the data leakage issue in Section 5.1, their efforts to verify the type and parameters of the generated results against user instructions do not fully mitigate this problem. Given that this benchmark is based on repositories from GitHub, and considering that nearly all large language models (LLMs) utilize data from GitHub, the static nature of these benchmarks may render this work less effective in the future. Since the paper has already implemented a sandbox environment for executing these repositories, a suggestion is to make the benchmark dynamic, similar to what LiveCodeBench has done. This could be achieved by regularly updating the benchmark with new repositories, thereby mitigating data leakage issues more effectively.

2. Enhanced Experimental Settings: To provide a clearer experimental analysis, it would be beneficial to include two additional experimental settings. In Table 4, the paper compares different levels of context for generation, similar to the approach in RepoBench. To enhance this analysis, experiments should be conducted with a more advanced retriever, such as UniXCoder, and without any extra context. The first addition would help determine whether a better retriever can lead to improved performance, while the second would allow for a deeper analysis of data leakage issues.

3. Table 4 lacks the latest SOTA LLMs, such as Claude-3.5-Sonnet and CodeQwen2.5. These models have demonstrated superior coding capabilities, both in closed and open-source environments. It would be insightful to evaluate whether these advanced LLMs could achieve superior performance on this benchmark.

**Questions:**

Check Weaknesses.

---

> ### Author Response · Authors · 2024-11-23
> **Weakness 1**
>
> We sincerely thank the reviewer for their thorough and constructive feedback. We particularly appreciate the recognition of our benchmark's comprehensiveness, thorough experimental analysis, and reproducible sandbox environment—elements that the reviewer notes make this "**a high-quality paper**" suitable for a "**top-tier conference**."
>
> ---
> > Although the paper addresses the data leakage issue in Section 5.1, their efforts to verify the type and parameters of the generated results against user instructions do not fully mitigate this problem. Given that this benchmark is based on repositories from GitHub, and considering that nearly all large language models (LLMs) utilize data from GitHub, the static nature of these benchmarks may render this work less effective in the future. Since the paper has already implemented a sandbox environment for executing these repositories, a suggestion is to make the benchmark dynamic, similar to what LiveCodeBench has done. This could be achieved by regularly updating the benchmark with new repositories, thereby mitigating data leakage issues more effectively.
>
> We appreciate this insightful concern regarding data leakage. **To address this critical issue comprehensively:**
>
> 1. We have introduced a new experimental setting where models are provided with only instructions, without any repository information. The results demonstrate a significant performance degradation across all models in this "zero-shot" setting (e.g., Claude-3.5-Sonnet drops from 23.85% Pass@5 to 5%, GPT-4o from 24.62% to 11.54%). This substantial performance gap empirically validates that models genuinely require repository information rather than merely memorizing previously encountered code.
>
>
> | Model               | Setting         | Pass@1 | Pass@2 | Pass@5 |
> |---------------------|-----------------|---------|---------|---------|
> | Claude-3.5-Sonnet   | Full Code       | 20.00   | 21.54   | 23.85   |
> |                     | BM25 Retrieval  | 11.54   | 15.38   | 19.23   |
> |                     | Oracle          | 25.38   | 27.69   | 30.00   |
> |                     | No Context      | 5.00    | 6.92    | 11.15   |
> | Claude-3.5-Haiku    | Full Code       | 16.92   | 19.62   | 21.54   |
> |                     | BM25 Retrieval  | 8.85    | 10.38   | 12.31   |
> |                     | Oracle          | 20.38   | 22.69   | 23.84   |
> |                     | No Context      | 2.69    | 4.23    | 4.23    |
> | OpenAI GPT-4O-Mini  | Full Code       | 21.15   | 22.69   | 24.62   |
> |                     | BM25 Retrieval  | 15.77   | 21.15   | 24.62   |
> |                     | Oracle          | 38.85   | 44.62   | 50.00   |
> |                     | No Context      | 11.54   | 14.62   | 18.08   |
> | OpenAI GPT-4O       | Full Code       | 18.46   | 20.00   | 23.08   |
> |                     | BM25 Retrieval  | 12.69   | 14.62   | 19.23   |
> |                     | Oracle          | 43.46   | 51.92   | 61.15   |
> |                     | No Context      | 11.54   | 14.61   | 19.23   |
> | Qwen2.5-7B-Instruct | Full Code       | 12.31   | 15.38   | 18.08   |
> |                     | BM25 Retrieval  | 11.92   | 16.92   | 19.38   |
> |                     | Oracle          | 33.46   | 39.62   | 47.31   |
> |                     | No Context      | 10.00   | 15.00   | 20.00   |
> | Qwen2.5-72B-Instruct| Full Code       | 17.31   | 18.46   | 20.38   |
> |                     | BM25 Retrieval  | 12.69   | 17.69   | 21.54   |
> |                     | Oracle          | 38.08   | 44.62   | 47.69   |
> |                     | No Context      | 11.92   | 17.31   | 19.62   |
> | Qwen2.5-Coder-32B   | Full Code       | 15.00   | 16.54   | 19.23   |
> |                     | BM25 Retrieval  | 22.31   | 26.54   | 32.31   |
> |                     | Oracle          | 40.38   | 44.92   | 51.92   |
> |                     | No Context      | 14.32   | 16.15   | 19.23   |
>
>
> 2. To further investigate data leakage impacts, we evaluated state-of-the-art models with recent training cutoffs (e.g., Qwen2.5 series, trained through 2024). Their suboptimal performance on ML-Bench indicates that even with potential data leakage, the inherent complexity of repository-level understanding presents significant challenges.
>
>
> 3. Regarding the dynamic benchmark suggestion, while we agree this would be ideal, the current benchmark relies on human annotation for quality assurance. Automated annotation methods for complex repository-level tasks remain an open challenge. We would be eager to incorporate automatic updates once reliable automated annotation methods become available. But we still propose an approach:
> Implementing an automated pipeline for periodic repository updates
> Developing semi-automated annotation tools to reduce manual labeling costs

---

> > ### Author Response · Authors · 2024-11-23
> >
> > | Model                                      | Setting         | Pass@1 (Full Code) | Pass@1 (No Context) | Delta |
> > |-------------------------------------------|-----------------|----------------------------|-----------------------------|-----------------------|
> > | Claude-3.5-Sonnet                         | Full vs No Context | 20.00                      | 5.00                        | 15.00                 |
> > | Claude-3.5-Haiku                          | Full vs No Context | 16.92                      | 2.69                        | 14.23                 |
> > | OpenAI GPT-4o-Mini                        | Full vs No Context | 21.15                      | 11.54                       | 9.61                  |
> > | OpenAI GPT-4o                             | Full vs No Context | 18.46                      | 11.54                       | 6.92                  |
> > | Meta-Llama-3.1-8B-Instruct-Turbo          | Full vs No Context | 5.77                       | 1.92                        | 3.85                  |
> > | Meta-Llama-3.1-70B-Instruct-Turbo         | Full vs No Context | 19.23                      | 1.15                        | 18.08                 |
> > | Meta-Llama-3.1-405B-Instruct-Turbo        | Full vs No Context | 13.46                      | 0.77                        | 12.69                 |
> > | DeepSeek-Chat                             | Full vs No Context | 10.38                      | 1.92                        | 8.46                  |
> > | DeepSeek-Coder                            | Full vs No Context | 12.31                      | 1.15                        | 11.16                 |
> >
> > Here, the performance gap between Oracle and No-context settings quantifies the potential effects of data leakage.
> > Even with potential exposure to training data, state-of-the-art models struggle, highlighting the genuine challenge posed by repository-level understanding tasks.

---

> > > ### Comment · Reviewer_5PFk · 2024-11-26
> > > **Response**
> > >
> > > The authors' responses address my concerns, and I will keep my score unchanged.

---

> ### Author Response · Authors · 2024-11-23
> **Weaknesses 2-3**
>
> > **Enhanced Experimental Settings:**
> To provide a clearer experimental analysis, it would be beneficial to include two additional experimental settings. In Table 4, the paper compares different levels of context for generation, similar to the approach in RepoBench. To enhance this analysis, experiments should be conducted with a more advanced retriever, such as UniXCoder, and without any extra context. The first addition would help determine whether a better retriever can lead to improved performance, while the second would allow for a deeper analysis of data leakage issues.
>
> We appreciate these constructive suggestions. **In response:**
>
> 1. **Inclusion of No-Context Setting:**
>    - We have implemented the "no-context" setting in the above table, as suggested. The **stark performance drop in this setting validates the importance of repository understanding.**
>
> Our experimental design now presents a comprehensive difficulty gradient:
> - task1 (Oracle): Theoretical performance ceiling with human-annotated context
> - task2 (Complete code): Real-world application scenario
> - task3 (BM25 retrieval): Intermediate complexity reduction
> - task4 (No context): Zero-shot baseline
>
> 2. **Retriever-Based Experiments:**
>    - The performance differential between BM25 retrieval (Task 3) and full code access (Task 2) is notable, highlighting significant potential for retrieval optimization.
>    - For example, **GPT-4O improves from 19.23% to 61.15% Pass@5** when provided full code access instead of BM25 retrieval.
>
> 3. **Future Exploration of Advanced Retrievers:**
>    - While we acknowledge that more advanced retrievers like UniXCoder exist, we deliberately chose BM25 as our baseline retriever for its widespread adoption and proven effectiveness
>    - Our framework is modular by design, enabling the community to develop and integrate more sophisticated retrieval modules
>    - We view this as an opportunity for community contribution rather than a limitation
>
> ---
>
> > Table 4 lacks the latest SOTA LLMs, such as Claude-3.5-Sonnet and CodeQwen2.5. These models have demonstrated superior coding capabilities, both in closed and open-source environments. It would be insightful to evaluate whether these advanced LLMs could achieve superior performance on this benchmark.
>
> We have expanded our evaluation to include current SOTA models:
>
> 1. **Claude-3.5 Series:**
>    - Sonnet: Achieves **30.00% Pass@5** in Oracle setting.
>    - Haiku: Achieves **23.84% Pass@5** in Oracle setting.
>
> 2. **Qwen2.5 Series:**
>    - Qwen2.5-72B-Instruct: Achieves **47.69% Pass@5** in Oracle setting.
>    - Qwen2.5-Coder-32B: Achieves **51.92% Pass@5** in Oracle setting.
>
> | Model               | Setting         | Pass@1 | Pass@2 | Pass@5 |
> |---------------------|-----------------|---------|---------|---------|
> | Claude-3.5-Sonnet   | Full Code       | 20.00   | 21.54   | 23.85   |
> |                     | BM25 Retrieval  | 11.54   | 15.38   | 19.23   |
> |                     | Oracle          | 25.38   | 27.69   | 30.00   |
> | Claude-3.5-Haiku    | Full Code       | 16.92   | 19.62   | 21.54   |
> |                     | BM25 Retrieval  | 8.85    | 10.38   | 12.31   |
> |                     | Oracle          | 20.38   | 22.69   | 23.84   |
> | OpenAI GPT-4o-Mini  | Full Code       | 21.15   | 22.69   | 24.62   |
> |                     | BM25 Retrieval  | 15.77   | 21.15   | 24.62   |
> |                     | Oracle          | 38.85   | 44.62   | 50.00   |
> | OpenAI GPT-4o       | Full Code       | 18.46   | 20.00   | 23.08   |
> |                     | BM25 Retrieval  | 12.69   | 14.62   | 19.23   |
> |                     | Oracle          | 43.46   | 51.92   | 61.15   |
> | Qwen2.5-7B-Instruct | Full Code       | 12.31   | 15.38   | 18.08   |
> |                     | BM25 Retrieval  | 11.92   | 16.92   | 19.38   |
> |                     | Oracle          | 33.46   | 39.62   | 47.31   |
> | Qwen2.5-72B-Instruct| Full Code       | 17.31   | 18.46   | 20.38   |
> |                     | BM25 Retrieval  | 12.69   | 17.69   | 21.54   |
> |                     | Oracle          | 38.08   | 44.62   | 47.69   |
> | Qwen2.5-Coder-32B   | Full Code       | 15.00   | 16.54   | 19.23   |
> |                     | BM25 Retrieval  | 22.31   | 26.54   | 32.31   |
> |                     | Oracle          | 40.38   | 44.92   | 51.92   |
>
>
>
> These results yield several **important insights**  :
>    - Even state-of-the-art models face significant challenges in repository-level understanding
>    - Model scale and parameter count somehow correlate with performance improvement
>    - Code-specialized models (e.g., Qwen-Coder) demonstrate advantages in specific settings
>
> **These findings validate ML-Bench's utility as a discriminative benchmark and highlight opportunities for future model improvements.**

---

> ### Author Response · Authors · 2024-11-26
>
> Dear Reviewer ``5PFk``,
>
> We sincerely thank you for your positive and encouraging feedback on our submission. Your recognition of our benchmark’s comprehensiveness, experimental rigor, and reproducibility as being of "high quality" and suitable for a "top-tier conference" is deeply appreciated.
>
> Best regards,
>
> Authors

---

### Author Response · Authors · 2024-11-25
**General Response**

We greatly appreciate the reviewers’ recognition of the strengths and contributions of our work. Specifically, **Reviewers `5PFk`, `gnam`, and `R7MP` have unanimously highlighted our work as "high-quality" and "interesting to appear in a top-tier conference"**, further affirming the value of ML-BENCH.


---

#### Key Acknowledgments

- **Reviewer `5PFk`** praised our **comprehensive benchmarks**, **thorough experimental analysis**, and **reproducible sandbox environment**, recognizing the critical elements of our work that address real-world ML challenges.

- **Reviewer `gnam`** acknowledged our unique focus on **ML applications' dependencies and dataset preparation**, emphasizing the practical relevance of our benchmark.

-  **Reviewer `R7MP`** commended the novelty of our **end-to-end ML tasks** and recognized the potential value of our **dataset analyses across different settings**, despite noting areas for improvement in clarity and presentation.



---

#### Revisions Made

We have made significant revisions and additions to the manuscript to address the reviewers' feedback (**highlighted in red in the uploaded revision PDF**). Below is a summary of the major updates:

1. **Presentation Improvements**:
   - Per Reviewer `R7MP`’s feedback, we renamed *ML-LLM-Bench* and *ML-Agent-Bench* to **ML-Bench-L** and **ML-Bench-A**, eliminating confusion with other benchmarks.
   - We restructured the methodology and analysis sections for better logical flow, removed unnecessary notations, and relocated detailed technical discussions to appendices, improving clarity as suggested by Reviewer `R7MP`.
   - We refined our description of task challenges and clearly differentiated our work from similar benchmarks to emphasize our contributions to automated ML workflow evaluation.

2. **Expanded Experiments**:
   - At the request of Reviewer `5PFk`, we evaluated **state-of-the-art (SOTA) models**, including `Claude-3.5` and `Qwen2.5` series, with cutting-edge results (e.g., `Qwen2.5-Coder-32B` achieving 51.92% Pass@5).
   - As suggested by Reviewer `5PFk`,  we introduced **no-context baselines** to empirically validate the reliance on repository understanding over memorization. For instance, `Claude-3.5-Sonnet`'s Pass@5 performance dropped significantly from 23.85% to 5% in this setting.
   - We analyzed task completion requirements to show that **95% of tasks require deep code understanding**, as raised by Reviewer `KizE`. This statistic, supported by examples, emphasizes the benchmark’s emphasis on repository-level comprehension rather than superficial documentation usage.



3. **Methodology and Evaluation**:
   - Addressing Reviewer `gnam`’s concerns, we clarified success criteria for ML-Bench-A, focusing on **deterministic evaluation based on repository usage patterns** and API conformance rather than stochastic ML outcomes.
   - Evaluation reliability was ensured through multiple annotators (co-authors) with high agreement (Cohen's kappa = 0.92).
   - We explained the **test-train split strategy**, as requested by Reviewer `R7MP`, to avoid overlaps and ensure that test cases are distinct and diverse.


4. **Improved Accessibility**:
   - Per Reviewer `R7MP`, we ensured that the **complete dataset**, code, and evaluation scripts are accessible, with thorough documentation for inspection and reproduction.
   - In response to Reviewer `gnam`’s ethics concern regarding annotation, we clarified that the annotators are co-authors, and their contributions are fully acknowledged in the manuscript.

5. **Ethical Considerations**:
To address Reviewer `gnam`'s flagged concern, we explicitly clarified that the annotators involved in dataset construction are co-authors and were integral to the research process. This information is now highlighted in the revised manuscript.

---

We believe our revisions and the accompanying empirical evidence strongly address the reviewers' concerns while reinforcing the novelty and robustness of our contributions. The updates to methodology, experimental design, and accessibility ensure a more comprehensive and polished presentation of ML-Bench. We welcome further discussions if there are additional questions.


Best regards,
**Authors**

---

> ### Author Response · Authors · 2024-12-03
>
> We appreciate the reviewers for their time and constructive feedback. We would like to understand if our responses have addressed your concerns fully. Please let us know if there are any further questions or confusions.

---

### Meta-Review · Area_Chair_JGdx · 2024-12-21

**Metareview:**

The paper introduces ML-Bench, a benchmark designed to assess large language models in generating code for real-world machine learning applications using existing repositories. The ML-Bench Dataset comprises 9,641 annotated examples from 18 GitHub repositories, challenging LLMs to handle user-specified arguments and complex documentation. There are still some major concerns in this work: limited novelty due to similarity to previous repo-level code generation work such as SWEBench and RepoBench, potential data leakage due to public access to ML repos, and presentation issues due to technical naming and content organization.

**Additional Comments On Reviewer Discussion:**

The points raised by the reviewers:
- Novelty: Similarity to previous benchmarks (RepoBench, SWE-Bench) raises questions about originality.
- Data Leakage: Concerns about models memorizing GitHub repository data.
- Presentation: Issues with clarity, organization, and overuse of notations.
- Evaluation Metrics: Suggestions to include advanced retrievers and more state-of-the-art (SOTA) models like Claude-3.5 and Qwen2.5.
- Task Design: Bias towards bash script tasks and reliance on README-documented tasks questioned.

During the rebuttal, the authors addressed some of the concerns above. However, concerns about the benchmark tasks, novelty, and data leakage are not fully addressed, resulting in mixed evaluations.

---

### Decision · Program_Chairs · 2025-01-22

Reject